# Extensive site-directed mutagenesis reveals interconnected functional units in the alkaline phosphatase active site

**Fanny Sunden[1]\*, Ariana Peck[1], Julia Salzman[1], Susanne Ressl[2], Daniel Herschlag[1]\***

[1]Department of Biochemistry, Beckman Center, Stanford University, Stanford, United States; [2]Molecular and Cellular Biochemistry Department, Indiana University Bloomington, Bloomington, United States

**Abstract** Enzymes enable life by accelerating reaction rates to biological timescales. Conventional studies have focused on identifying the residues that have a direct involvement in an enzymatic reaction, but these so-called 'catalytic residues' are embedded in extensive interaction networks. Although fundamental to our understanding of enzyme function, evolution, and engineering, the properties of these networks have yet to be quantitatively and systematically explored. We dissected an interaction network of five residues in the active site of *Escherichia coli* alkaline phosphatase. Analysis of the complex catalytic interdependence of specific residues identified three energetically independent but structurally interconnected functional units with distinct modes of cooperativity. From an evolutionary perspective, this network is orders of magnitude more probable to arise than a fully cooperative network. From a functional perspective, new catalytic insights emerge. Further, such comprehensive energetic characterization will be necessary to benchmark the algorithms required to rationally engineer highly efficient enzymes.

**\*For correspondence:** fsunden@
stanford.edu (FS); herschla@
stanford.edu (DH)

**Competing interests:** The
authors declare that no
competing interests exist.

**Reviewing editor**: John Kuriyan,
Howard Hughes Medical
Institute, University of California,
Berkeley, United States

## Introduction

Scientists have long marveled at the enormous rate enhancements and exquisite specificities of enzymes. Remarkable progress has been made since catalysis was viewed as a 'life force' just over a century ago (*Buchner, 1897*; *Hein, 1961*; *Barnett, 2003*). Now, the chemical moieties involved in enzymatic transformations can be identified by a combination of structural and functional approaches. The positions of functional groups in X-ray structures can typically be combined with chemical intuition to derive mechanisms for enzymatic reactions (*Benkovic and Bruice, 1966*; *Walsh, 1979*; *Sinnott, 1998*; *Silverman, 2002*). Such mechanisms are routinely supported by site-directed mutagenesis experiments in which large deleterious rate effects are observed when putative catalytic residues are mutated.

While there remain a subset of reactions that are less understood and whose important and fascinating reaction details are still being worked out (e.g., *Das et al., 2011*; *Weeks et al., 2012*), we can write reasonable chemical mechanisms for the vast majority of enzymes (*Benkovic and Bruice, 1966*; *Walsh, 1979*; *Sinnott, 1998*; *Silverman, 2002*). In contrast, our understanding of the energetics that underlie enzymatic catalysis is far less developed. Such understanding is critical for elucidating the pathways that have been followed in molecular evolution and for designing new, highly efficient enzymes.

The current dominant mechanistic tools, X-ray crystallography and removal of catalytic residues via site-directed mutagenesis, while powerful, have fundamental limitations. Structures reveal the positions of functional groups in active sites, but these static pictures do not allow reaction probabilities to be determined. Specifically, although energies derived from a given structure can, in

**eLife digest** Enzymes are biological catalysts that speed up the reactions that are essential for life. As such, enzymes convert 'reactant' molecules into other molecules. Reactant molecules bind to part of the enzyme called the active site. Some of the amino acids that make up the active site must directly interact with these molecules to catalyze the reaction.

Mutating individual active site amino acids often greatly reduces or destroys the ability of the enzyme to increase reaction rates. These amino acids are known as 'catalytic residues'. However, catalytic residues do not work in isolation: instead, they interact with other residues in the enzyme to carry out their function. Therefore, the effects of these interactions need to be characterized in order to fully understand how enzymes work.

Sunden et al. explored the interactions within a network of five residues found at the active site of an enzyme, called alkaline phosphatase, which was taken from the bacterial species *E. coli*. Nearly all of the possible combinations of these five residues were examined. The results of these experiments indicated that even though all five residues are structurally linked, only a subset of the residues affected one another functionally, even though all of them are structurally connected. In particular, three groups—or functional units—of residues were found in the enzyme structure. The residues within each functional unit directly or indirectly cooperate to increase different aspects of the enzyme's catalytic activity. Sunden et al. used this information to develop models that describe how the functional units work together, and suggest that the likelihood of the active site evolving so that its residues are not fully cooperative is high.

It remains to be seen whether similar cooperative networks exist in the active sites of other enzymes and how residues further away affect those in and around the active site. Understanding how the residues in the active site work together and being able to model their interactions could help efforts to develop more efficient enzymes for use in biotechnology in the future.

principle, reveal potential energies, full sampling of the ensemble states of the enzyme, substrates, and surrounding solvent is needed to obtain free energies. It is difficult to combine such sampling with high-level energy functions and to make nontrivial predictions required for independent assessment of such energetic models.

Site-directed mutagenesis has two fundamental limitations. First, site-directed mutagenesis reads out the difference in free energy of the reaction ($\Delta\Delta G$) for the mutant enzyme relative to the wild type (WT), and so does not report an absolute energetic contribution to catalysis (*Kraut et al., 2003*; *Herschlag and Natarajan, 2013*). As an example, the vastly different assignments of the catalytic contributions of residues in the Ketosteroid Isomerase oxyanion hole, dependent on the type and extent of mutation, provide a particularly clear demonstration of this limitation and underscore the need to clearly and explicitly define the comparison states (*Kraut et al., 2010*; *Schwans et al., 2011*).

The second limitation of site-directed mutagenesis is that enzymatic residues do not act in isolation (*Narlikar and Herschlag, 1998*; *Kraut et al., 2003*). The apparent contribution of one residue is also a function of the surrounding residues and the overall structure, as demonstrated by energetic coupling in double mutant cycles and more dramatically by the fact that a denatured protein still contains its catalytic residues but these so-called catalytic residues no longer provide catalysis (see *Kraut et al., 2003* for discussion) (*Carter et al., 1984*; *Horovitz, 1996*).

One seemingly important aspect of the interconnectivity of enzymatic residues is highlighted in X-ray structures, which typically reveal or suggest active site hydrogen bond networks and imply a functional connection between the identified catalytic residues and these more extensive 'network residues'. Indeed, prior investigations have identified networks of energetically coupled residues in enzymes that contribute synergistically to catalysis. These studies include incisive double mutant cycle analysis demonstrating functional and energetic connections between residues (*Hermes et al., 1990*; *Dion et al., 1993*; *Horovitz et al., 1994*; *Rajagopalan et al., 2002*; *Masterson et al., 2008*; *Singh et al., 2014*). Larger scale coupled networks have been observed through statistical analysis of co-evolution and characterization of individual residue's relaxation timescales by NMR (e.g., *Lockless and Ranganathan, 1999*; *Eisenmesser et al., 2002, 2005*; *McElheny et al., 2005*; *Freedman et al., 2009*; *Halabi et al., 2009*; *Doucet et al., 2011*).

A remaining challenge, undertaken in this study, is to link mutant cycle analysis to extended networks. Currently, we lack the ability to ascertain the energetic properties of network residues from structural inspection, first principles, or empirical models. For example, do these networks act as fully cooperative units where disruption of any connection would dissipate the advantage from all of the residues in the network? In the other extreme, do certain side chains act independently from their network neighbors, positioned for function by their backbone placement and/or packing interactions with other portions of the side chain?

Although either extreme might be presumed unlikely, such expectations are not grounded in data given the current absence of quantitative assessments of these extended networks. We have therefore investigated the functional behaviors of an interaction network hypothesized from available structural data in the *Escherichia coli* alkaline phosphatase (AP) active site (*Figure 1*). Our experiments reveal and quantitatively delineate the energetic interconnectivity within this highly proficient active site, the type of active site that will be necessary to create if we are to engineer enzymes that rival the catalytic power of natural enzymes.

## Results and discussion

*Figure 1A* shows the three-dimensional structure of *E. coli* AP; for simplicity, a single monomer of the homodimeric active enzyme is presented. *Figure 1B,C* shows a close-up and a schematic depiction of the network of residues in the active site, respectively, and *Figure 1D* depicts the reaction catalyzed by AP. As expected based on structural data, the presence of the $Zn^{2+}$ ions and the active site nucleophile, S102, are required for measureable activity (*Plocke et al., 1962*; *O'Brien and Herschlag, 2001*; *Andrews et al., 2013*), and prior work has revealed functional effects from mutation of some of the $Zn^{2+}$ ligands (*Xu and Kantrowitz, 1992*; *Tibbits et al., 1994*, *1996*; *Ma et al., 1995*).

Here, we focus on the other residues in the active site and determine their functional and energetic connectivity (*Figure 1C*). The energetic effects we refer to here and in the remainder of this study are related to free energy differences. Prior work identified E322, which is required for $Mg^{2+}$ binding, and R166 as catalytic residues, with mutations of these leading to 88,000 and 6300-fold decreases in the rate of catalytic activity, respectively (*Zalatan et al., 2008*; *O'Brien et al., 2008*). Nevertheless, these residues are part of an extensive hydrogen-bonded and $Mg^{2+}$-coordinating network that also involves D101, D153, K328, the $Mg^{2+}$ ion liganded by E322, and two water molecules (*Figure 1C*).

As described in the following sections, we determined the effects of removing each of the five side chains of this apparent network individually and in combination. To determine if a residue's contribution is independent of or dependent on the other side chains, we determined the activity of a minimal form of the AP active site with all five residues mutated, herein referred to as AP minimal (D101A/D153A/R166S/E322Y/K328A), and we also restored each side chain individually to this minimal enzyme. As described below, the results indicated a strong interdependence of the residues, prompting us to explore these interconnections by making nearly all possible combinations of these five mutants (28 out of 32; $2^n$, where n = 5, the number of positions mutated). Our results identified three interconnected functional units, each with distinct underlying energetic properties, and allowed us to develop a quantitative model that reproduces the observed rate constants and predicts the rate constants of the remaining four mutants not tested herein. For ease of reference, all of the measured and calculated rate constants are listed in *Table 1* and shown graphically in *Figure 2A* and *Figure 2B*.

### Testing catalytic residues for independence vs interdependence

*Figure 3A* shows the effects from mutating each of the five active site residues depicted in *Figure 1C*. Each residue has a significant effect ranging from 64 to 88,000-fold, with the largest effects coming from R166 mutation and $Mg^{2+}$ ion removal (E322Y). Previous studies have shown that replacing the E322 side chain with a tyrosine leads to loss of $Mg^{2+}$ from the active site (*Zalatan et al., 2008*). We confirmed this result and showed that the E322Y mutant reaction is not activated by the presence of $Mg^{2+}$ ('Materials and methods'). The absence of $Mg^{2+}$ binding and activation is consistent with the finding that other members of the AP superfamily that lack $Mg^{2+}$ have a tyrosine at this position (*Zalatan et al., 2008*). Indeed, we chose the E322Y mutation for our studies because it gives the same functional effect as E322A while also creating this steric block to $Mg^{2+}$ binding (*Zalatan et al., 2008*). While we did not test all possible mutations, several different mutations were tested at each position and shown to give very similar effects, providing no indication of idiosyncratic effects from any of the subtractive mutations made in this study (*Appendix 1 Table 1*).

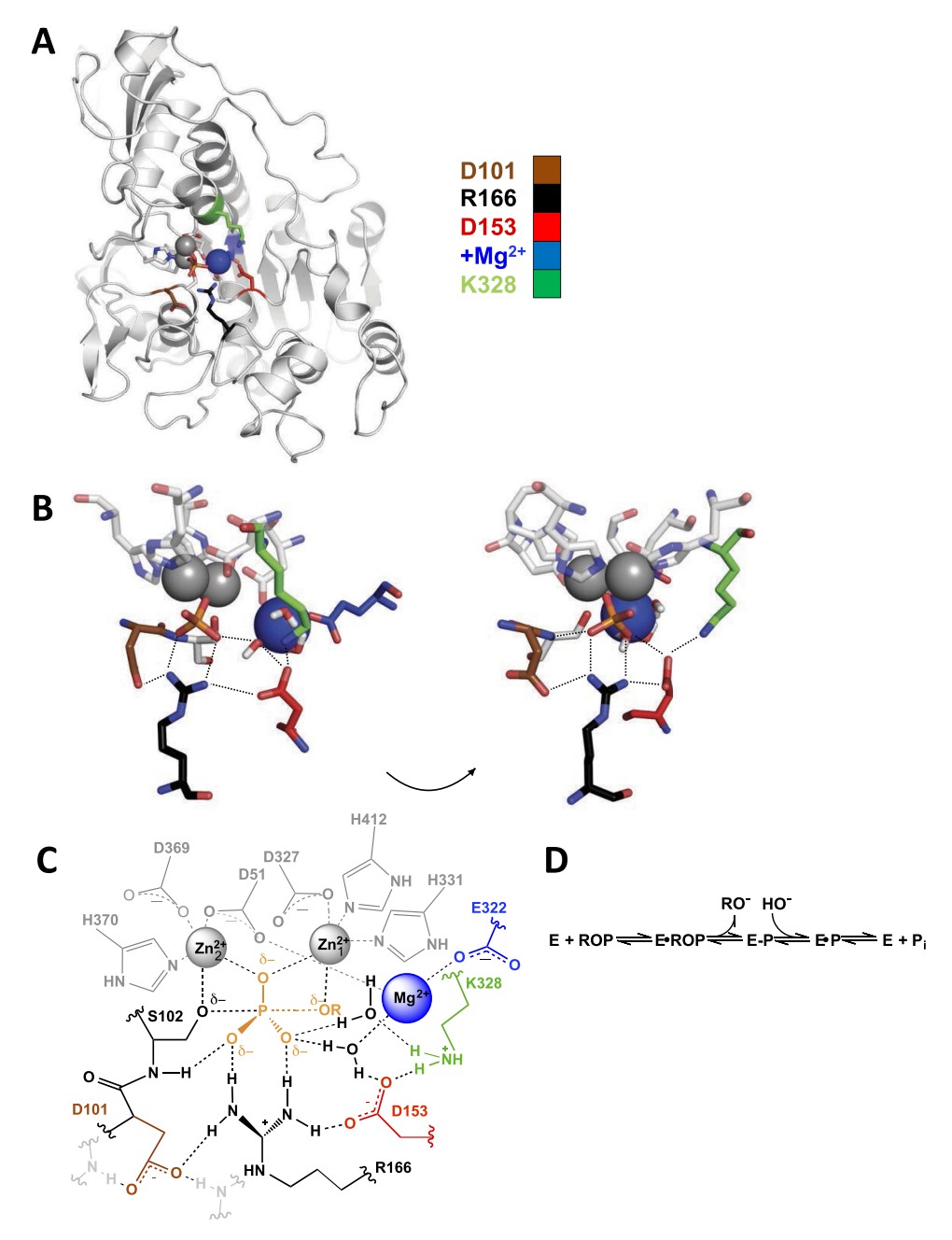

**Figure 1**. Alkaline phosphatase (AP) structure and active site. (**A**) The three-dimensional structure of AP with bound $P_i$ (PDB 3TG0). Active site residues are depicted as follows: D101, brown; R166, black; D153, red; K328, green; E322 and $Mg^{2+}$ ion, blue. (**B**) A close-up of AP active site from two angles. Dashes represent putative hydrogen bonds. Residues colored as in part (**A**). (**C**) Schematic of AP active site interactions represented with the phosphoryl transfer transition state. Residues colored as in part (**A**). (**D**) Reaction scheme for phosphomonoester hydrolysis by AP, where ROP represents a phosphate monoester dianion substrate, and E-P represents the covalent seryl-phosphate intermediate (**Coleman, 1992**).

We next determined the degree to which these residues were dependent on one another in two simple ways. First, we removed all five residues simultaneously; if independent, the effects of removal would be multiplicative—that is, energetically additive (**Equation 1**). However, the cumulative effect assuming additivity gives a predicted activity that is $10^6$-fold lower than the observed activity of the

**Table 1**. Kinetic constants for WT and mutant AP*,†

| AP mutant | $k_{cat}/K_M$ (M$^{-1}$s$^{-1}$) | fold decrease† | $K_M$ (µM) | $k_{cat}$ (s$^{-1}$) |
|---|---|---|---|---|
| WT | $3.3 \times 10^7$‡ | – | $3.6 \times 10^{-1}$ | 12 |
| | $(6.3 \times 10^8)$§ | (1) | | |
| D101A | $9.9\ (2.0) \times 10^6$ | 64 | 3.6 (0.9) | 36 (9) |
| R166S¶,# | $1.0 \times 10^5$ | $6.3 \times 10^3$ | 5.0 | 0.5 |
| D153A | $2.8\ (0.4) \times 10^6$ | $2.3 \times 10^2$ | 2.6 (0.7) | 7.6 (2.8) |
| E332Y**,# | $7.2\ (2.2) \times 10^3$ | $8.8 \times 10^4$ | ~0.5 | – |
| K328A# | $7.5\ (2.4) \times 10^5$ | $8.4 \times 10^2$ | 5.4 (1.8) | 3.4 (0.9) |
| D101A/R166S | $5.8\ (0.2) \times 10^4$ | $1.1 \times 10^4$ | 8.4 (2.3) | $4.2\ (2.0) \times 10^{-1}$ |
| D101A/D153A# | $3.3\ (0.7) \times 10^5$ | $1.9 \times 10^3$ | 6.2 (2.1) | 2.2 (0.7) |
| D101A/E322Y# | 3.1 (0.1) | $2.0 \times 10^8$ | $1.5\ (0.3) \times 10^2$ | $4.6\ (0.8) \times 10^{-4}$ |
| D101A/K328A†† | $(2.8 \times 10^4)$ | $2.3 \times 10^4$ | – | – |
| R166S/D153A | $1.3\ (0.1) \times 10^4$ | $4.9 \times 10^4$ | 5.3 | $7.0 \times 10^{-2}$ |
| R166S/E322Y** | 1.6 (0.5) | $3.9 \times 10^8$ | 27 (11) | – |
| R166S/K328A# | $2.4\ (0.4) \times 10^2$ | $2.6 \times 10^6$ | $6.3\ (2.5) \times 10^1$ | $1.6\ (0.4) \times 10^{-2}$ |
| D153A/E322Y | $2.3\ (0.1) \times 10^3$ | $2.7 \times 10^5$ | $5.2\ (0.4) \times 10^{-1}$ | $1.4\ (0.3) \times 10^{-3}$ |
| D153A/K328A | $4.4\ (1.4) \times 10^5$ | $1.4 \times 10^3$ | 2.4 | 1.1 |
| E322Y/K328A | $1.5\ (0.1) \times 10^3$ | $4.2 \times 10^5$ | 2.2 | $3.4 \times 10^{-3}$ |
| D101A/R166S/D153A# | $2.0\ (0.7) \times 10^4$ | $3.2 \times 10^4$ | 4.8 (1.4) | $8.3\ (1.8) \times 10^{-2}$ |
| D101A/R166S/E322Y†† | $(4.2 \times 10^{-2})$ | $1.5 \times 10^{10}$ | – | – |
| D101A/R166S/K328A†† | $(2.8 \times 10^2)$ | $2.3 \times 10^6$ | – | – |
| D101A/D153A/E322Y†† | $(1.3 \times 10^1)$ | $4.8 \times 10^7$ | – | – |
| D101A/D153A/K328A | $1.2\ (0.1) \times 10^5$ | $5.3 \times 10^3$ | 11 | 1.6 |
| D101A/E322Y/K328A | 1.1 (0.1) | $5.7 \times 10^8$ | 78 | $8.9 \times 10^{-5}$ |
| R166S/D153A/E322Y | $1.9\ (0.1) \times 10^1$ | $3.3 \times 10^7$ | $4.4\ (0.2) \times 10^2$ | $8.3\ (1.0) \times 10^{-3}$ |
| R166S/D153A/K328A | $3.6\ (0.1) \times 10^3$ | $1.8 \times 10^5$ | 7.2 (0.1) | $2.6\ (0.1) \times 10^{-2}$ |
| R166S/E322Y/K328A | $3.9\ (0.8) \times 10^{-1}$ | $1.6 \times 10^9$ | $1.4\ (0.3) \times 10^2$ | $5.7\ (1.1) \times 10^{-5}$ |
| D153A/E322Y/K328A | $3.2\ (0.3) \times 10^2$ | $2.0 \times 10^6$ | 2.1 (0.3) | $7.3\ (0.3) \times 10^{-4}$ |
| D101A/R166S/D153A/E322Y | $9.4\ (1.6) \times 10^{-1}$ | $6.7 \times 10^8$ | $3.5 \times 10^2$ | $3.2\ (0.7) \times 10^{-4}$ |
| D101A/R166S/D153A/K328A | $5.5\ (1.7) \times 10^3$ | $1.2 \times 10^5$ | 12 (5.0) | $6.1\ (0.3) \times 10^{-2}$ |
| D101A/R166S/E322Y/K328A | $\leq 2.0 \times 10^{-2}$ | $\geq 3 \times 10^{10}$ | – | – |
| D101A/D153A/E322Y/K328A | 2.0 (0.1) | $3.2 \times 10^8$ | $1.4\ (0.1) \times 10^2$ | $2.4\ (0.3) \times 10^{-4}$ |
| R166S/D153A/E322Y/K328A# | 4.4 (0.1) | $1.4 \times 10^8$ | $2.2\ (0.1) \times 10^2$ | $9.6\ (0.5) \times 10^{-4}$ |
| D101A/R166S/D153A/E322Y/K328A# | $1.7\ (0.3) \times 10^{-1}$ | $3.7 \times 10^9$ | $1.6\ (0.6) \times 10^2$ | $2.9\ (0.7) \times 10^{-5}$ |

*Activities were measured in 0.1 M MOPS, pH 8.0, 0.5 M NaCl, 100 µM ZnCl$_2$, and 500 µM MgCl$_2$ at 25°C. Errors in activities are standard deviations from duplicate with the same or independent enzyme preparations (see 'Materials and methods').

†'Fold down' values are $k_{cat}/K_M$ for WT AP estimated for reaction with the chemical step rate limiting (footnote d and description in Appendix 1) divided by ($k_{cat}/K_M$) for each mutant. By definition the value for WT AP is one.

‡($k_{cat}/K_M$)$_{obsd}$ (**O'Brien and Herschlag, 2002**).

§For WT AP, the chemical step is not rate limiting (**O'Brien and Herschlag, 2002**). As has been carried out previously, we used comparisons with a substrate for which the chemical step is rate limiting to estimate the value of $k_{cat}/K_M$ for WT AP that would be expected with fast association and the chemical step rate limiting (see description in Appendix 1). Relative values are compared to this number.

#The mutant was expressed in two independent enzyme preparations and standard deviations are from activity measurements for the independent preparations.

¶From reference (**O'Brien et al., 2008**).

**From reference (*Zalatan et al., 2008*).
††Values in brackets were not measured but are calculated from the energetic behavior of the functional units according to the mathematical model described in Appendix 1 and shown in *Appendix 1 Table 2*.

minimal construct (*Equation 1*), indicating substantial energetic interdependence between these residues (*Figure 3A*);

$$\Delta\Delta G^{\ddagger} = \left[ -RT\ln\left( (k_{cat}/K_M)_{\mathbf{obsd}} \big/ (k_{cat}/K_M)_{\mathbf{predicted}} \right) \right] = 8.3\ \text{kcal/mol},$$

$$k_{rel,}^{predicted} = k_{rel,\ D101} \times k_{rel,\ R166} \times k_{rel,\ D153} \times k_{rel,\ K328} \times k_{rel,\ Mg^{2+}}. \tag{1}$$

We next added each residue back in isolation to the minimal AP construct lacking all five residues, and we compared the rate effect from the added residue to the corresponding rate decrease from removing that residue from WT AP (*Figure 3A,B*). In each case, there was a larger deleterious effect from removing the residue than the rate enhancement afforded by adding it back. The differential effects ranged from 2.6 to >1900-fold, corresponding to differential energetic effects of up to at least 4.5 kcal/mol. Indeed, D153 added back in isolation is deleterious by at least 10-fold (*Figure 3B*) but beneficial by 230-fold when added in the otherwise WT background (*Figure 3A*). Further, the small 2.6-fold (0.57 kcal/mol) differential between restoring D101 to AP minimal and removing it from WT AP likely arises coincidentally from different mechanistic contributions (see below). Thus, each of the five residues has some energetic connection to at least one residue in this network. Overall, the contribution from adding all five residues back is 580-fold greater than predicted from assuming energetically additive effects from addition of each residue individually (*Equation 1*).

An empirical or theoretical framework to explain this energetic behavior is lacking. In particular, there is no way to discern from structural inspection of an active site whether all neighbors have substantial energetic interactions, the scale and form of these energetic connections, or whether and how far beyond nearest neighbors functional effects extend. Therefore, we proceeded to define the interconnections between active site residues for this model enzyme system.

## Deep mutagenesis to uncover the interrelationships between active site residues: interconnected catalytic units

*Figure 2A* depicts the set of all 32 possible combinations of mutations of the five catalytic residues, using the color-coding from *Figure 1* to represent the presence (colored) or absence (white) of each residue. In the following sections, we describe the energetic properties and interconnections of these residues.

### The D101/R166/D153 functional unit

R166 was previously identified as a catalytic residue in the AP active site and hypothesized to stabilize the transition state via hydrogen bonds to the non-bridging oxygens (*Figure 1C*) (*O'Brien and Herschlag, 1999*; *O'Brien et al., 2008*). Mutation of R166 to serine or alanine has a similar energetic effect (*O'Brien et al., 2008*), and given the energetic similarity of R166S and R166A and the practical limits on the number of mutants that could be investigated, we used R166S in all subsequent experiments.

The effects of the other residues on R166 were determined by comparing the effect of mutation of this residue (R166S) in 13 different mutant backgrounds. We first determined the effects of D153 and D101, which are both within hydrogen bonding distance of R166. In the extremes, if the function of R166 was fully dependent on the neighboring aspartate residues then there would be little or no effect from adding back the R166 side chain in the absence of both D101 and D153. Alternatively, if R166 was independent of its neighbors, its full catalytic effect would be restored regardless of the presence of the aspartate residues.

Relative to the full effect of 6300-fold from addition of the R166 side chain with all other residues present, addition of this side chain in the absence of both aspartate side chains gives a much smaller rate increase of only ~10-fold (*Figure 4*, left black bar). The presence of either aspartate residue increases the catalytic effectiveness of R166, such that reintroduction of the arginine side chain contributes 170 or 220-fold with D153 or D101 present, respectively, more than the increase from

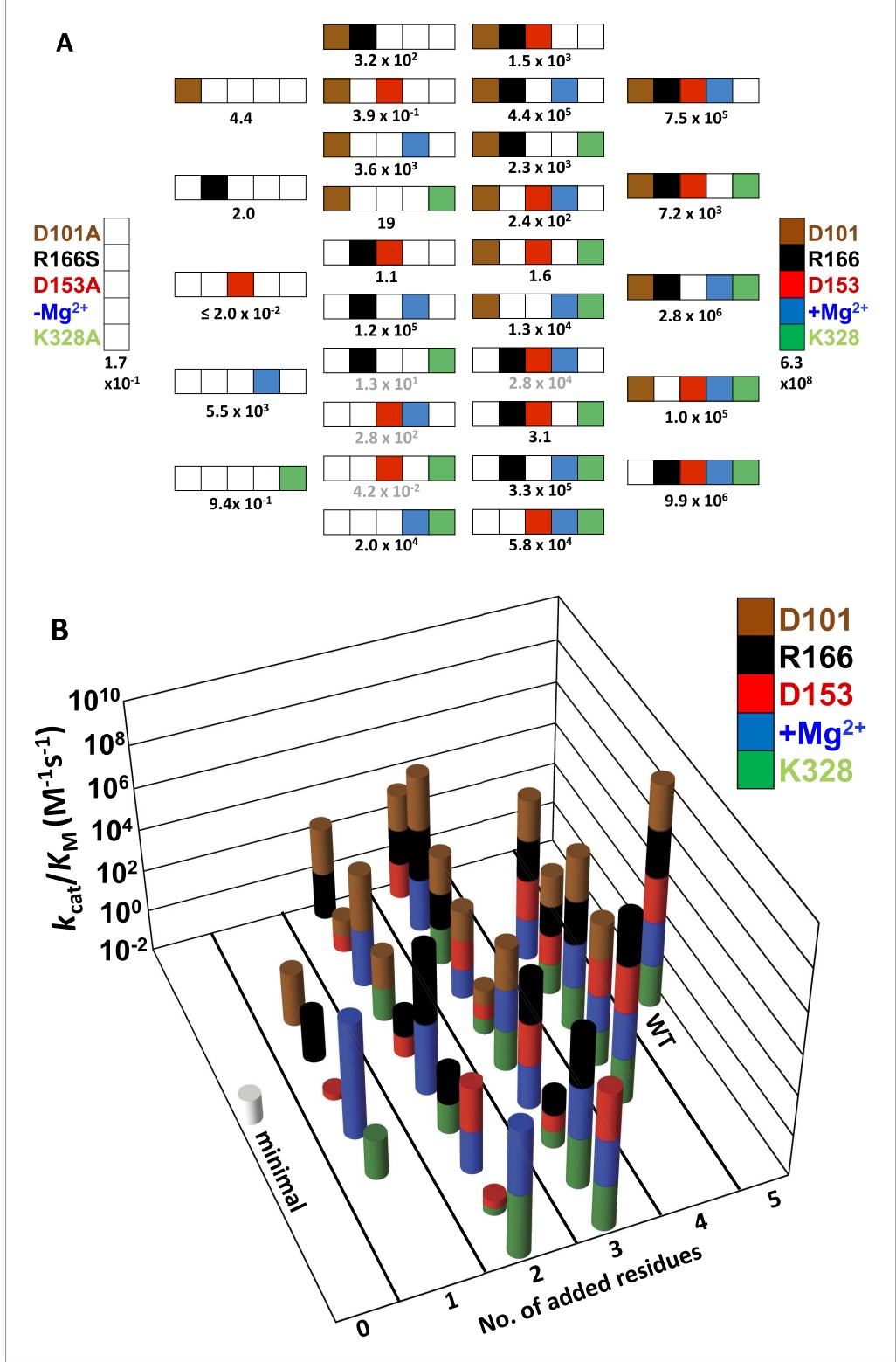

**Figure 2**. Catalytic efficiencies of AP variants for all combinations of five active site residues. (**A**) The 32 possible combinations of the five residues diagrammed with color-coding of residues as in *Figure 1* to represent whether a particular WT residue is present: D101, brown; R166, black; D153, red; K328, green; and E322 and the $Mg^{2+}$ ion, blue; the absence of a WT residue at a particular position is indicated by a white square. The catalytic efficiency, $k_{cat}/K_M$ ($M^{-1}s^{-1}$), of each combination is noted below each construct (*Table 1*). Rate constants calculated from the

*Figure 2. continued on next page*

*Figure 2. Continued*
energetic behavior of the functional units are shown in grey (*Table 1*). (**B**) Three-dimensional representation of the activities of the 32 AP variants, with the height of each bar corresponding to $k_{cat}/K_M$ ($M^{-1}s^{-1}$ on a log scale) and the same color scheme as in (**A**).

R166 addition with neither aspartate residue present. However, this is still considerably less than the 6300-fold increase with both aspartate residues present (*Figure 4*, black bars).

These results indicate that the function of R166 is interconnected with and enhanced by its neighbors, D101 and D153. The simplest model to account for these effects is that each aspartate residue helps to position the arginine side chain for interaction with the non-bridging phosphoryl oxygen atoms (*Figure 1C* and see below).

We next asked whether the other network side chains are needed for full R166 function and enhance its function through either or both aspartate residues. The gray bars in *Figure 4* compare the effects from reintroduction of the arginine side chain with or without the neighboring aspartate residues as in the black bars from this figure described above, but now comparing the effects in the presence or absence of K328 and the active site $Mg^{2+}$. The effect of arginine add-back is nearly the same with and without K328 and the $Mg^{2+}$ ion (cf. the pairs of black and gray bars in *Figure 4*). Hence, R166, D101, and D153 form a functional unit, independent of the $Mg^{2+}$ ion and K328.

## Structural effects of D101 and D153

Based on the functional effects shown in *Figure 4*, we expected mutation of D101 and D153 to result in a misaligned or less-ordered R166. This expectation is supported by prior X-ray structures of single mutants of D101 (*Chen et al., 1992*) or D153 (PDB 1AJC, 1AJD) and by a structure of the D101A/D153A double mutant collected as part of this investigation.

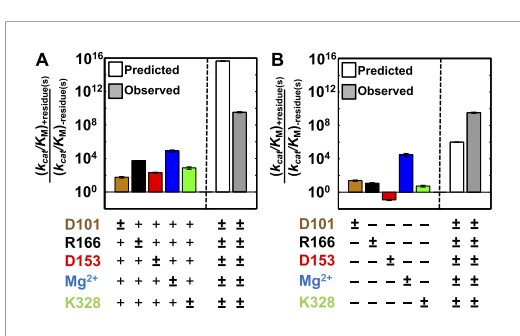

**Figure 3**. Single-mutation effects and additivity predictions. Rate effects from removing individual residues from WT AP (**A**) or restoring individual WT residues to AP minimal (**B**). The symbol (±) indicates which residue is varied. Residues are color-coded as in *Figure 1*: D101, brown; R166, black; D153, red; K328, green; and E322 and $Mg^{2+}$ ion, blue. The following mutations were made: D101A, R166S, D153A, K328A, and E322Y; several alternative mutations gave similar effects (*Appendix 1 Table 1*). To the right of the dashed line is the activity of WT AP relative to AP minimal observed (**A**, **B**, grey bars) and predicted from the effects of removal of each WT residue from the WT background and assuming independent (energetically additive) effects (**A**, open bar) or from the effects of addition of each WT residue in the minimal background, assuming independence (**B**, open bar).

R166 adopts the same position in structures of WT AP, regardless of the presence of $P_i$ in the active site (PDB 1ED9, 3TG0). Independent structures of apo D153G AP (PDB 1AJC, 1AJD) show displacement of R166 relative to its WT position. In one of these structures, the guanidinium group of R166 is rotated by ~19°, which would distort the hydrogen bond angles with the non-bridging oxygen atoms of a phosphate bound as it is in WT AP (*Figure 5—figure supplement 1*). In the second structure of D153G AP, the R166 side chain is flipped away from the active site. The crystal structure of apo D101S similarly shows displacement of R166 (*Chen et al., 1992*). In contrast, E322Y AP, which lacks the bound $Mg^{2+}$ ion that is not part of the R166 functional unit, does not influence the position of R166 (*Zalatan et al., 2008*; *Figure 5—figure supplement 2*).

To further explore the effects of the aspartate residues on the R166 functional unit, we solved the crystal structure of D101A/D153A AP (*Table 2*). In one subunit of the AP homodimer, the guanidinium group is flipped away from the active site and adopts a catalytically unproductive conformation that is similar to the conformation observed in one structure of D153G, but more displaced than in the other structure of D153G and D101S AP (*Figure 5A*). In the other

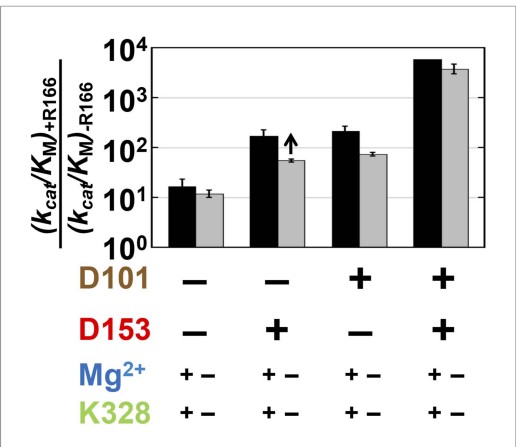

**Figure 4**. Catalytic effects of R166 in different mutant backgrounds. The effects of restoring R166 in different aspartate backgrounds with the Mg²⁺ ion and K328 present (black) or absent (grey). The arrow above the bar in the D153 background indicates that the ratio is a lower limit. Residues are color-coded as in *Figure 1*, rate constants are from *Table 1*, and mutations made are listed in *Table 1*.

monomer, R166 was observed to populate two rotameric positions, one of which is similarly flipped away from the active site while the other is pointed towards the active site. (*Figure 5B*). Although this structure lacks a bound Mg²⁺ ion in the active site, the functional and structural data presented above suggest that this loss will not influence the position of R166. In both active sites of D101A/D153A AP, the bound $P_i$ is not well positioned, populating multiple conformations that are each distinct from its position in WT AP, which likely closely mimics the reactive phosphoryl conformation (Appendix 2 and *Appendix 2 Figure 1*) (*Holtz et al., 1999*; *O'Brien et al., 2008*). These structural comparisons provide evidence that the aspartate residues that flank R166 help to correctly position the reactive phosphoryl group and provide a self-consistent picture of the functional effects of the D101/R166/D153 functional unit. Nevertheless, a future challenge will be to quantitatively link structural rearrangements and resulting structural ensembles to functional effects (*Frederick et al., 2007*).

## Possible roles of groups beyond the D101/R166/D153 functional unit in R166 positioning

Single hydrogen bonds between the aspartate residues and R166 might be expected to precisely position the arginine residue only if other interactions position the aspartate residues themselves. We have determined that the remaining groups of the apparent active site network, K328 and the Mg²⁺ ion, are not required, so positioning of the aspartate side chains must arise from distinct interactions.

The D101 carboxylate group accepts a hydrogen bond from a backbone amide and therefore could be directed by that interaction (*Figure 1C*). In contrast, the D153 carboxylate group does not appear to accept a hydrogen bond from residues beyond the five probed herein; however, its methylene group packs against the side chains of nearby residues.

Testing the role of the backbone amide in positioning D101 to, in turn, position R166 will be difficult, but the model for D153 is readily testable: mutation of the nearby side chains is predicted to eliminate the ability of D153 to stimulate R166 function. It will also be of interest to determine if, in the absence of the side chains that putatively pack around and position D153, the hydrogen bond network that comprises K328, the Mg²⁺ ion, and their associated water molecules is now needed to position D153 and, in turn, position the R166 side chain.

### The D153/Mg²⁺(E322)/K328 functional unit

The Mg²⁺ ion in the AP active site had previously been identified as important for catalysis, decreasing activity ~10⁵-fold when removed by replacing one of the metal ligands, E322, with a tyrosine or alanine, an effect even larger than that from removal of R166 (*Zalatan et al., 2008*). We assessed possible functional connections to the Mg²⁺ ion analogously to the approach taken above for R166.

*Figure 6* shows the effect from addition of the bound Mg²⁺ ion (via restoration of E322; *Zalatan et al., 2008* and 'Materials and methods') in eight different mutant backgrounds. These comparisons test the energetic interconnectivity of the Mg²⁺ ion with all residues other than D101, and we address D101 and the Mg²⁺ ion in the following section ('The D101/Mg²⁺(E322) Functional Unit'). There is a substantial rate increase of 820-fold from Mg²⁺ ion addition in the absence of any of the other three residues (*Figure 6*, far left). This effect is ~100-fold less than the maximal effect of 88,000-fold from Mg²⁺ ion addition in the otherwise WT context (*Figure 6*, far right).

Consistent with the results presented above for R166, the presence of R166 has virtually no effect on the rate advantage from adding Mg²⁺ back to the active site, regardless of which other residues

**Table 2.** X-ray crystallographic data collection and refinement statistics

| Data collection | |
|---|---|
| Space group | P6₃22 |
| Unit cell axes | |
| $a, b, c$ (Å) | 161.4, 161.4, 140.1 |
| $\alpha, \beta, \gamma$ (°) | 90.0, 90.0, 120.0 |
| Resolution range (Å) | 52.89–2.24 (2.31–2.24)*,† |
| $R_{merge}$ (%) | 43.0 (305.9) |
| $R_{pim}$ (%)‡ | 9.6 (71.9) |
| $<I>/<\sigma I>$ | 7.8 (1.4) |
| Completeness (%) | 100.0 (100.0) |
| Multiplicity | 21.6 (19.7) |
| $CC_{1/2}$ | 0.996 (0.582) |
| Refinement | |
| Resolution range (Å) | 52.89–2.24† |
| No. unique reflections | 51,819 (5086) |
| $R_{work}/R_{free}$ (%) | 21.7/25.9 |
| Number of atoms | 6794 |
| Average B-factors (Å²) | |
| protein | 37.8 |
| water | 30.2 |
| ligands | 41.2 |
| R.m.s. deviation from ideality | |
| Bond length (Å) | 0.006 |
| Bond angles (°) | 1.10 |
| Ramachandran statistics | |
| Favored regions (%) | 98.4 |
| Allowed regions (%) | 1.6 |
| Outliers (%) | 0 |
| PDB code | 4YR1 |

*Values in parenthesis are for the highest resolution shell.
†The high resolution cut-off applied during scaling and refinement was decided based on $CC_{1/2}$ and completeness (**Diederichs and Karplus, 2013**; **Evans and Murshudov, 2013**).
‡$R_{pim}$ is reported in addition to $R_{merge}$ due to the high multiplicity of the data set.

are present or absent (**Figure 6**, mutant sets −/+R166). Also, when either D153 or K328 is present, the effect from $Mg^{2+}$ add-back remains the same. Only when both D153 and K328 are present is the full 88,000 effect observed (**Figure 6**, far right). The ~100-fold differential effect with or without D153 and K328 indicates that the $Mg^{2+}$ ion, D153, and K328 form a unit that is functionally separate from the catalytic contributions of the D101/R166/D153 unit.

Nevertheless, the two functional units are interconnected as they have D153 in common. From a functional perspective, given the close quarters within an active site, even if fully independent functional units were readily obtainable, interconnected units might still be preferable to allow a compact active site that can accommodate the density of groups needed to provide multiple interactions with centrally located substrates and thus optimal catalysis. From an evolutionary perspective, parsimony in natural selection might be expected to lead to multiple uses of the same residue, such that one residue could serve in two functional units, and one functional unit might serve as a foundation or scaffold from which to build other functional units, thereby leading to interconnected functional units. This idea is related to a proposal of Hanson and Rose that natural selection favors the use of the minimal number of acidic and basic residues in active sites, so that such residues tend to be reused when mechanistically reasonable (**Hanson and Rose, 1975**).

Whereas AP in its fully evolved form has functionally distinct D101/R166/D153 and D153/$Mg^{2+}$/K328 units (i.e., these functional units do not provide an additional advantage to one another), mutations that disrupt the surrounding AP scaffold could alter these energetics and possibly render these units energetically interdependent (**Narlikar and Herschlag, 1998**; **Kraut et al., 2003**). Such mutations may have been present during the early evolution of the AP active site, when its scaffold may have been less precisely arranged.

## The D101/$Mg^{2+}$(E322) functional unit

Given the absence of direct connections between the active site $Mg^{2+}$ or K328 and D101 and the above findings of circumscribed functional units, we initially expected that there would be no interactions between D101 and these groups. Data analogous to that of **Figures 4, 6** probing the dependence of the K328 contribution with or without D101 confirmed this expectation for K328 (**Figure 8—figure supplement 1**). However, a connection was observed between D101 and the $Mg^{2+}$ ion. Addition of D101 when the $Mg^{2+}$ ion is not present leads to a ~20-fold rate increase (**Figure 7A**); in contrast, there is no increase when $Mg^{2+}$ is present (**Figure 7B**). **Figure 7A,B** shows only cases in which R166 is not present because D101 enhances the potency of R166 as part of the D101/R166/D153 functional unit (**Figure 4**); nevertheless, the interconnection of D101 with R166 and the $Mg^{2+}$ ion are distinct, as the enhancement by D101 of

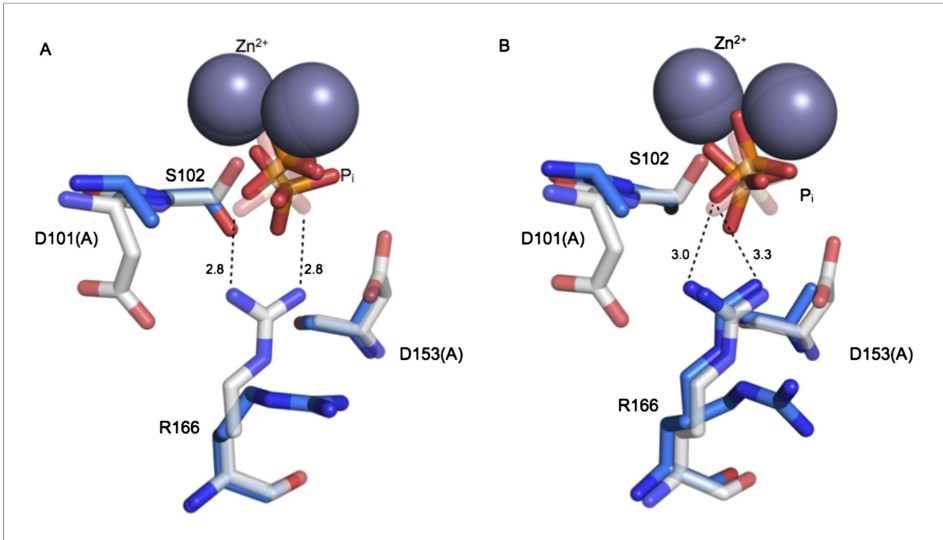

**Figure 5**. Structure of the active site of D101A/D153A AP. The active sites of the superimposed crystal structures of
$P_i$-bound D101A/D153A AP (protein: blue, $P_i$: orange) and $P_i$-bound WT AP (protein: white, $P_i$: transparent). The two
monomers of the AP dimer exhibit different active site configurations and are therefore both shown (**A**, **B**). (In
contrast, the WT AP monomers are remarkably similar, as can be seen by comparing panels **A** and **B**.) In both
monomers of D101A/D153A AP, $P_i$ populates two positions that are distinct from its position in WT AP. (**A**) In one
active site of D101A/D153A, R166 is rotated away from the active site. The hydrogen bond distances between R166
and $P_i$ in WT AP are 2.8 Å (shown in **A**). (**B**) In the other active site of D101A/D153A, R166 partially occupies two
positions, one of which faces the active site and hydrogen bonds to one of the partially occupied $P_i$ ions. The other
rotameric position adopted by R166 is flipped away from the active site, and presumably coincides with the other
partially occupied $P_i$ ion, as it would sterically clash with the active-site facing R166 rotamer. Hydrogen bond
distances and angles for WT and D101A/D153A AP are listed in *Appendix 2 Table 1*.

The following figure supplements are available for figure 5:

**Figure supplement 1**. Ablation of D153 disrupts R166 positioning.

**Figure supplement 2**. Removal of the active site $Mg^{2+}$ ion does not disrupt R166 positioning.

the R166 contribution is the same with and without the $Mg^{2+}$ ion present (*Figure 4*). Together, these
results indicate that D101 has a stimulatory effect independent of R166, but only when the $Mg^{2+}$ ion is
not present.

To further explore coupling between D101 and the $Mg^{2+}$ ion, we plotted the effect of adding the
$Mg^{2+}$ ion with or without D101 present (*Figure 8*). The effect from the added $Mg^{2+}$ ion varies
depending on whether D153 and K328 are present, as described in the preceding section, but in each
case the effect is diminished if D101 is present. Mirroring the larger effect from D101 addition in the
absence of $Mg^{2+}$, addition of the $Mg^{2+}$ ion had a 30-fold larger effect in the absence of D101.

Thus, D101 and the $Mg^{2+}$ ion form a third functional unit, with energetic behavior distinct from the
other two units. In the D101/R166/D153 functional unit, R166 has close to energetically additive effects
from the neighboring aspartate residues, and the $Mg^{2+}$ functional unit (D153/$Mg^{2+}$/K328) exhibits
cooperative energetic effects from K328 and D153. In contrast, the $Mg^{2+}$ ion and D101 are energetically
anti-cooperative, with a 20–30-fold contribution from each occurring only when the other group is absent.

These observations raise two questions addressed immediately below: (1) how are D101 and the
$Mg^{2+}$ ion functionally linked, and (2) what is the origin of their anti-cooperative energetics?

## How are D101 and the $Mg^{2+}$ ion functionally linked?

Consideration of the AP active site structural network reveals potential linkages between D101 and
the $Mg^{2+}$ ion. They are linked via R166 and D153 (*Figure 1C*), but these residues have no effect on the
energetic coupling between D101 and the $Mg^{2+}$ ion, so models involving these residues and this

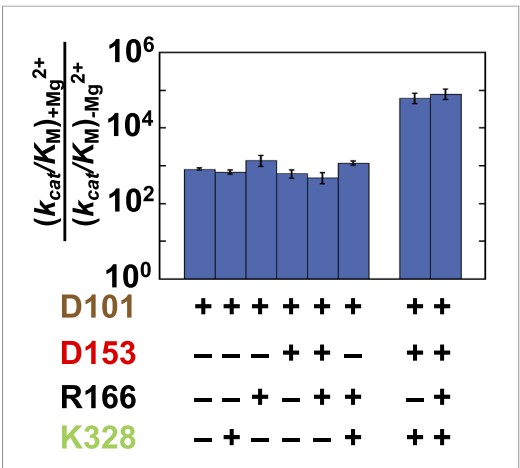

**Figure 6**. Catalytic effects of the Mg²⁺ ion in different mutant backgrounds. The effect of Mg²⁺ ion addition in different mutational backgrounds. Residues are color-coded as in *Figure 1*, rate constants are from *Table 1*, and mutations made are listed in *Table 1*.

connection are ruled out. A second structural connection between D101 and the Mg²⁺ ion can be traced as follows: D51 is a ligand of both the Mg²⁺ ion and the Zn²⁺ ion that coordinates the nucleophilic oxyanion of S102 ($Zn_2^{2+}$), and D101 is directly upstream of S102, with its side chain anchored to the backbone amide of S102 (*Figure 1C*).

## What is the origin of their anti-cooperative energetics?

Given this connection, why are these energetically anti-cooperative—that is, how is a favorable effect from each residue prevented or obviated by the presence of the other residue? A model that D101 and the Mg²⁺ ion can each make a catalytic interaction alone that is prevented by a steric block or conformational rearrangement upon addition of the other residue would require fortuitous catalytic interactions present in mutant enzyme forms that are not maintained in WT and is therefore unlikely. A more appealing model is the one in which the two residues have a redundant effect, such that either residue can alone provide a rate advantage through the same mechanism. Given that D51 is a ligand of both the Mg²⁺ ion and the Zn²⁺ ion that coordinates the nucleophilic oxyanion of S102 ($Zn_2^{2+}$, *Figure 1B,C*), the Mg²⁺ ion could help position the serine oxyanion with respect to the rest of the catalytic apparatus (*Figure 1C*). D101 is directly upstream of S102 with its side chain anchored to the backbone amide of the S102. Thus, it is possible that either D101 or Mg²⁺ interactions can facilitate proper positioning of the S102 oxyanion for nucleophilic attack and that, once either of the two interactions is made, positioning is optimized such that no further functional enhancement occurs upon addition of the second (*Figure 1C*).

## Active site redundancy

As described above, our energetic data indicate that there is a redundant effect of the Mg²⁺ ion and D101. Redundancy is often observed in functional studies from the molecular to the cellular to the organismal level (*Peracchi et al., 1998*; *Naor et al., 2005*), but its origins are less clear. It is sometimes suggested that more 'important' pathways or functions evolve redundancy, but more complex mechanisms are required to maintain redundancy, otherwise it would simply be lost over evolution (*Cookea et al., 1997*). One intriguing idea is that redundancy is built to aid evolvability, and evolvability itself is a property that has been selected for over evolution (*Maynard Smith, 1978*; *Conrad, 1979*; *Joyce, 1997*). It is also possible that many reported cases of redundancy have individual functional effects that are too small to be detected in laboratory assays or have favorable functional effects that manifest only under natural growth conditions.

We propose a distinct origin of the D101/Mg²⁺ redundancy, arising from the high degree of connectivity within the AP active site. D101 and the Mg²⁺ ion are each involved in other functional

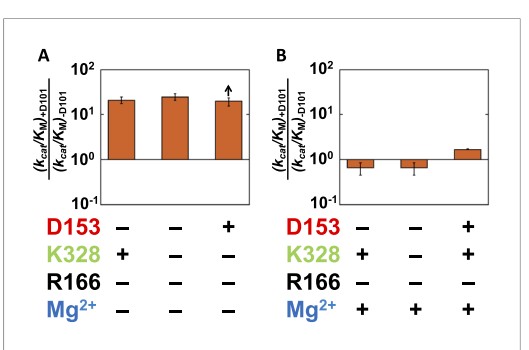

**Figure 7**. Catalytic effects of D101 in different mutant backgrounds. The effects of restoring D101 in backgrounds without bound Mg²⁺ (**A**) and with bound Mg²⁺ (**B**). The arrow indicates that the ratio is a lower limit. Residues are color-coded as in *Figure 1*, rate constants are from *Table 1*, and mutations made are listed in *Table 1*. R166 is absent because it is also coupled with D101 (*Figure 4*).

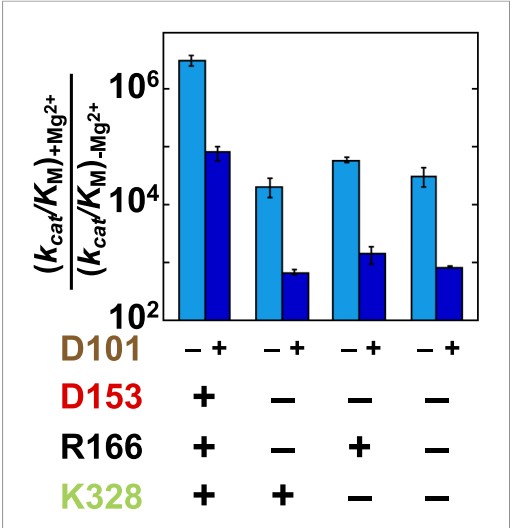

**Figure 8**. Catalytic effects of Mg$^{2+}$ ion removal in mutant backgrounds without and with D101 present. The effects of restoring Mg$^{2+}$ in the absence of D101 (light blue) and the presence of D101 (dark blue). Residues are color coded as in **Figure 1**, rate constants are from **Table 1**, and mutations made are listed in **Table 1**.

The following figure supplement is available for figure 8:

**Figure supplement 1**. Effect of K328 addition with (light green) or without D101 (dark green) present.

networks and are embedded in an extensively interwoven active site. These interconnections presumably position functional groups to make optimal catalytic interactions. As noted above, the functional networks overlap with one another that is, have shared residues (**Figure 9**), and these overlaps may have arisen as evolution co-opted residues within active sites as anchor points for positioning new residues. Given the high density of functional groups within active sites and the presumed need for multiple interconnections to provide optimal positioning of functional groups for transition state interactions, residues that evolved for different functional purposes may end up structurally linked and with the ability to position the same group or groups.

## Conclusions and implications

X-ray crystallographic structures reveal connections between residues but cannot define the energetic properties of these connections. Site-directed mutagenesis reveals residues that give large functional effects upon mutation, and double mutant cycles provide powerful tests for energetic dependence vs independence between two residues (**Carter et al., 1984**; **Hertel et al., 1994**; **Horovitz, 1996**; **Narlikar et al., 1999**). However, residues never function in isolation, and essentially every residue in a protein structure can be connected, via chains of hydrogen bonding and packing interactions, to every other residue. In cases of allostery, there are energetic interactions over large distances, and in certain cases the residues and conformations involved in the allosteric transition have been mapped (**Monod et al., 1965**; **Shulman et al., 2004**; **Cui and Karplus, 2008**; **McLaughlin et al., 2012**). We are particularly interested in active site networks because the residues constituting these networks are critical for the most basic catalytic functions of enzymes, these networks have not been previously mapped, and an understanding of their extent and properties may facilitate engineering of enzymes that rival natural enzymes in catalytic efficiency and specificity.

### Active site networks and functional units

We have interrogated a network of five residues in the active site of the model enzyme AP (**Figure 1B,C**; **Figure 2A**). A priori, these residues could have functioned fully cooperatively or fully independently. Although either extreme might have been considered unlikely, there were no data for this system (or, to our knowledge, other systems) that would allow us to know or predict the degree and extent of functional interconnectivity.

We observed three overlapping functional units, D101/R166/D153, D153/Mg$^{2+}$/K328, and D101/Mg$^{2+}$ (**Figure 9**). Each functional unit exhibited distinct energetic behaviors. The aspartate residues of the D101/R166/D153 unit make nearly independent contributions to the catalytic function of the central arginine residue; D153 and K328 (and presumably their associated water molecules) act cooperatively with the Mg$^{2+}$ ion and increase its contribution to catalysis; and the Mg$^{2+}$ ion and D101 are energetically anti-cooperative and apparently constitute a redundant functional unit.

A mathematical model that describes these energetics quantitatively accounts for all of the AP variants, reproducing the observed catalytic rates for all 28 mutants within a factor of two and predicting the catalytic rates of the remaining four variants that were not tested herein (**Appendix 1 Table 2**). Nevertheless, the atomic-level interactions and properties that underlie each of these

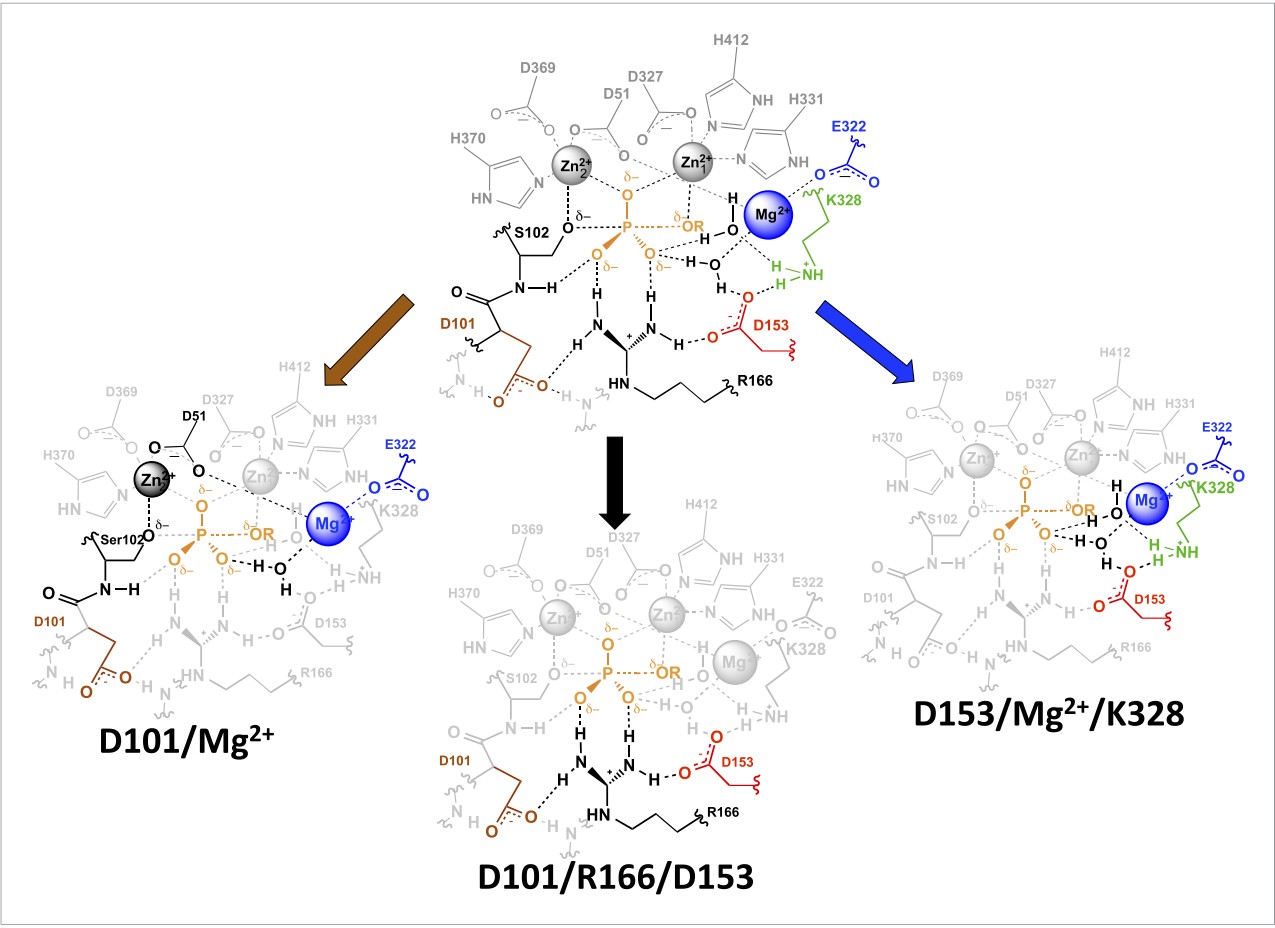

**Figure 9**. AP active site functional units. The residues of the functional unit are color-coded as in *Figure 1*. For the D101/Mg²⁺ functional unit (left), the black residues and Zn²⁺ ion represent a potential route for the energetic connections between these residues.

distinct and complex energetic behaviors are not known. We propose the following models: each aspartate residue of the D101/R166/D153 unit helps to position R166, thereby lowering the conformational entropy cost for making its interactions with the two phosphoryl oxygen atoms; the network of the D153/Mg²⁺/K328 functional unit and its two associated water molecules exists in alternative, likely less structured conformations until all three of these groups are present and can position hydrogen bond donors to one of the oxygen atoms of the transferred phosphoryl group; and D101 and the Mg²⁺ ion can each position the serine oxyanion nucleophile with respect to $Zn_2^{2+}$, rendering their contributions redundant. These models will require future testing and refinement.

The interconnections between the functional units (*Figure 9*)—that is, the residues common to more than one functional unit—may be favored both evolutionarily and functionally. An interconnected active site will have fewer residues and thus require a less expansive search of mutational space. Further, multiple functional groups need to 'pack' into a limited space to interact with the substrate and stabilize the transition state.

## Synergy between experimental and computational enzymology

The approach taken in this study introduces a powerful opportunity to deepen the feedback loop between experiment and computation. It is generally recognized that the most powerful tests of computational approaches are true predictions, made 'blindly' with pertinent data not yet collected or obtained by others but not released prior to reporting computational results and predictions. Such blind predictions have proven critical in other areas of biochemistry and biophysics to determine whether seemingly predictive and descriptive algorithms were indeed predictive and correct

(*Moult et al., 2004*, *2011*; *Nielsen et al., 2011*; *Cruz et al., 2012*; *Dill and MacCallum, 2012*). Nevertheless, the vast majority of computational studies of enzyme mechanism make 'predictions' for results that are already known and are thus not true independent tests. Further, single active site mutations predominantly give effects that fall within typical and rather narrow ranges, limiting the usefulness of traditional site-directed mutagenesis as a robust test of computational approaches.

We suggest that computational predictions of the rate effects from multiple mutations may provide extensive and nontrivial predictions that can be quantitatively tested by experiment and that are needed to effectively advance computational methods and our understanding. Our group is willing to test predictions from individual computational groups or consortia, using AP or other systems where we know robust kinetic measurements are possible; we are willing to carry out such experiments in advance and withhold the results or send them to an independent evaluator to ensure that comparisons between experiment and computation will be possible and made in a timely manner; and we are willing to discuss with computational groups the best systems to carry out such tests. We strongly believe that such synergistic approaches, informal and formal, will be required to unite computational and experimental enzymology and to make the greatest advances in our understanding of catalysis.

## Enzyme design

There has been recent excitement about the ability to design new enzymes, some of which catalyze reactions not seen in nature (*Röthlisberger et al., 2008*; *Siegel et al., 2010*; *Hilvert, 2013*; *Bos and Roelfes, 2014*). The ability to repurpose and create protein scaffolds and to place functional groups in desired locations is a truly remarkable advance. Nevertheless, designed enzymes to-date have modest rate enhancements relative to naturally occurring enzymes and can lack the stereospecificity observed with naturally occurring enzymes (*Wolfenden and Snide, 2001*; *Lassila et al., 2009*; *Baker, 2010*; *Bos and Roelfes, 2014*). Indeed, enzyme mimics and bovine serum albumin (BSA) can catalyze reactions with rate enhancements similar to designed enzymes prior to their improvement by randomization and selection (*Kirkby et al., 2000*; *Schmidt et al., 2013*).

These observations raise the following questions: what distinguishes naturally occurring enzymes from current designed enzymes, and what is needed to achieve more proficient designs? The most apparent difference is the absence of extended and extensive hydrogen bond networks in and around active sites of designed enzymes. For example, the most carefully studied designed enzyme, a retroaldolase, has a lysine residue placed within a hydrophobic pocket (*Jiang et al., 2008*). Rate enhancements are achieved by lowering the lysine p$K_a$ due to its non-polar environment (to increase the concentration of the reactive free amine at neutral pH) and from binding the hydrophobic substrate in this lysine-containing pocket; these mechanisms provide $\sim$10$^5$-fold catalysis for a designed enzyme, with modest additional rate enhancement obtained through subsequent rounds of selection (*Lassila et al., 2009*; *Hilvert, 2013*).

We hope that an empirical understanding of the extent and properties of active site networks will help in future design efforts as well as promote more conceptual and theoretical understanding. Although it would not be appropriate to generalize from the single example of AP, our dissection of the AP active site network does show that optimal positioning of catalytic residues can occur with only a subset of the full network; thus, it may be possible to attempt designs with active site modules that correspond to functional units identified in this study and, we hope, subsequent studies. A recent attempt to incorporate an active site catalytic triad produced designed enzymes that, after several rounds of selection, were able to catalyze a side reaction with efficiencies similar to analogous natural enzymes (*Rajagopalan et al., 2014*). Taking a longer view, the hand-in-hand development and testing of computational approaches, as outlined above (see 'Synergy between experimental and computational enzymology'), will ultimately provide foundational models for the efficient design of highly effective and specific new enzymes.

## Evolutionary pathways and probabilities

The evolution of a fully cooperative network would be exceedingly improbable and more difficult the larger the number of constituent residues, as all of the residues would need to arise through random drift with a selective advantage accruing only once the entire network was in place. A probability model in which each of five residues would arise with a one in twenty probability (i.e., one active residue out of the total 20 amino acids) and all are needed for a selective advantage (i.e., full

cooperativity) has a probability of $1/(6.8 \times 10^5)$ (*Figure 10A*, bottom pathway and Appendix 3). This probability arises because there are five steps (or positions) each with 20 possible residues, and there are five chances of 'choosing' a WT residue, starting with AP minimal with all five WT residues missing, four chances of choosing the next residue, etc.

Conversely, evolution of a hypothetical enzyme in which each active site residue provides an independent rate advantage would be considerably more probable, as addition of those residues would lead to monotonically increasing catalysis (i.e., be continually uphill on a fitness landscape and thus selected for after each addition). We first consider a simplified model with a single pathway of five successive steps that each lead to increased fitness, each with a one in twenty probability representing a single advantageous residue of the twenty possible residues (*Figure 10A*, top path, black numbers). The probability of achieving the final state for a multi-step process can be considered in terms of a mean waiting time, akin to a reaction's half-time or the inverse of its rate constant (scaled by ln 2). In this case, with five successive irreversible steps each with probability 1/20, the 'rate constant' is 1/100 and the mean wait time is 69 (in arbitrary units; Appendix 3). (Calculating probabilistic waiting times using a hidden Markov Model, such as is often used for probability calculations, gives the same relative values as the kinetic mean times [Appendix 3].) This kinetic mean wait time is about three orders of magnitude lower than that for the fully cooperative process, which is obtained by computing the mean waiting time: $4.7 \times 10^5$ ($=6.8 \times 10^5 \times$ ln 2; Appendix 3). Thus, if it were functionally equally probable to obtain an active site with these different underlying energetic properties (i.e., fully cooperative vs stepwise; *Figure 10A*), evolution would favor the non-cooperative solution by greater than 1000-fold. In other words, for every enzyme that evolved five residues with full functional dependence on one another, there would be more than 1000 with active sites containing residues whose stepwise addition each provided a selective advantage.

As we observed functional units exhibiting a range of energetic behaviors within the AP active site, we asked the question: where does the energetic behavior of AP place it along this wide range of evolutionary probabilities? There are 120 possible pathways from AP minimal to WT AP (*Figure 10B*). To assess the range of possible mean waiting times, we consider two limiting cases (*Figure 10A*, top pathway): i. The case in which there is a single pathway of the 120 that is favorable; this is the simplified model presented above and gives a mean wait time of 69. ii. The case in which all 120 potential pathways are favorable—that is, proceed with a selective advantage at each step; in this case the mean waiting time is 32, even shorter than for case (i) because there are more ways to traverse the landscape from AP minimal to WT AP.

Because we have rate constants for each AP species (*Table 1*; *Figure 2*), we can determine which pathways would confer a selective advantage. Using an arbitrary minimal cutoff of >threefold increase in $k_{cat}/K_M$, 34 of the 120 pathways confer a fitness advantage; with a cut-off of a fivefold increase in $k_{cat}/K_M$, 28 of these pathways remain. Thus, although the AP active site residues studied herein have varied energetic behaviors, in many cases addition of a WT residue leads to a significant rate increase, providing multiple favorable evolutionary routes. While not all 120 pathways are favorable for AP, many are, and the mean waiting time will thus be within the range of 32–69, >1000-fold shorter than the expected waiting time to evolve a fully cooperative network.

Although a fully cooperative network would probably not have been anticipated, we were unaware of how strong the evolutionary pressure would be for stepwise increases in fitness. Similar conclusions have been drawn for the evolution of unlinked loci where largely additive rather than fully cooperative or synergistic effects have been observed (*Arnegard et al., 2014*). It will be fascinating to explore more broadly the interplay of functional effects and evolutionary probabilities and how this interplay has biased the complement of extant enzymes. Such underlying probabilistic preferences presumably also impact biological solutions at higher levels of function such as gene regulation and neuronal function (*McLean et al., 2011*).

## Implications for exclusion of water from active sites

Many enzymes have loops or flaps that close over active sites or domains that accomplish analogous closure (*Pai et al., 1977*; *Bennett and Steitz, 1978*; *Alber et al., 1987*). These events exclude solvent, and it is often stated or implied that the exclusion of water provides the underlying driving force to evolve these processes (*Pai et al., 1977*; *Bennett and Steitz, 1978*; *Harris et al., 1997*; *Cleland et al., 1998*; *Richard and Amyes, 2004*). However, the observation that water is excluded does not logically indicate that the exclusion of water causes enhanced catalysis.

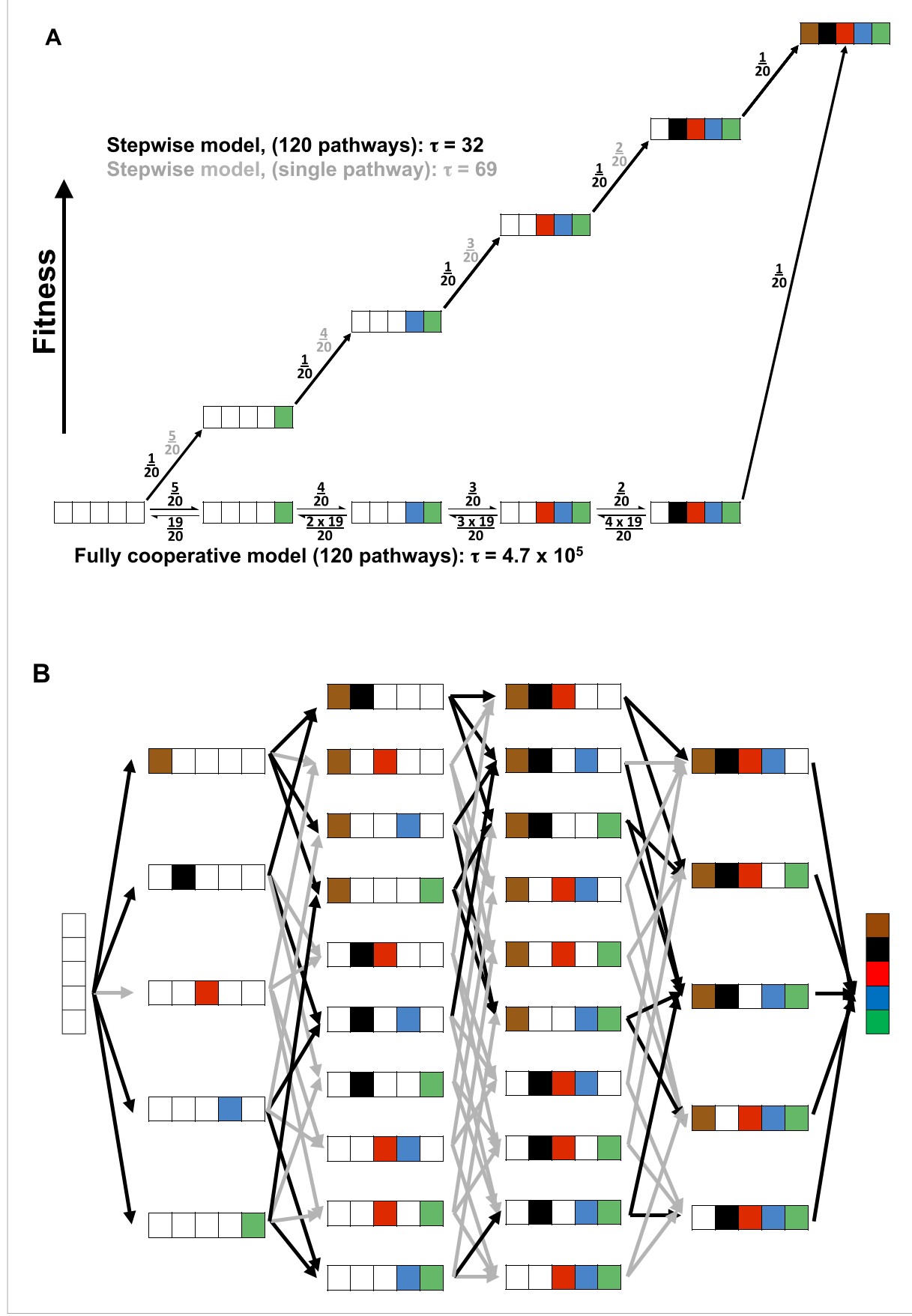

**A**

**Stepwise model, (120 pathways): τ = 32**
**Stepwise model, (single pathway):** τ = 69

Fitness

**Fully cooperative model (120 pathways): τ = 4.7 x 10⁵**

**B**

*Figure 10. continued on next page*

*Figure 10. Continued*

**Figure 10**. Cooperative and independent models of active site evolution. (**A**) Schematic comparing fully cooperative (bottom) and stepwise (top) models for a single pathway. In the fully cooperative model, simultaneous acquisition of all five WT residues is required to confer a selective advantage, leading to a mean waiting time of $4.7 \times 10^5$ (arbitrary units) considering all 120 pathways of adding in the residues (black net rates; for simplicity, only one intermediate of the multiple possible mutant combinations is shown in each step). In contrast, the stepwise model, in which acquisition of any WT residue confers a fitness advantage and is thus irreversible (top, black numbers), has a minimum mean waiting time of 32. If only one of the 120 pathways leads to a stepwise increase in fitness (top, grey numbers) then the mean waiting time would be 69. The model and simplifying assumptions made to highlight the differences arising from the presence or absence of cooperativity are described in Appendix 3. (**B**) Model of active site evolution showing the 120 possible paths in the AP landscape for introduction of the five residues investigated herein, in an otherwise WT background. A stepwise model in which acquisition of any WT residue is considered irreversible and all paths are possible would result in a mean waiting time of 32 (all arrows, grey and black, same as part **A**, top). As a subset of mutagenic steps toward WT AP (36 of the 80 potential evolutionary steps) confers a selective advantage (here defined as a rate increase of >threefold) and paths containing steps that do not confer such an advantage have much lower probabilities, we consider the 34 of 120 pathways that provide a monotonic fitness increase as all five WT residues are added. This gives a mean waiting time between the mean waiting times for the stepwise models for a single pathway and 120 pathways, 32 and 69, respectively.

AP has an open cavity to allow cleavage of a wide range of phosphate esters, consistent with its putative role in scavenging inorganic phosphate; nevertheless, it has one of the largest known rate enhancements, suggesting that full exclusion of water is not required to attain extremely efficient enzymatic catalysis.

AP does, however, appear to exclude water from access to the non-bridging phosphoryl oxygen atoms, and we considered whether that exclusion is catalytically important. Removal of R166 (and D101), which allows solvent access to the phosphoryl oxygen atoms (*O'Brien et al., 2008*) does not diminish the catalytic contribution from the $Mg^{2+}$ ion or the $D153/Mg^{2+}/K328$ functional unit, which also interact with at least one of these phosphoryl oxygen atoms. Thus, at least for AP, the wholesale exclusion of solvent from the active site does not enhance catalytic contributions. We suggest, most generally, that flap or domain closure allows the establishment of additional specific interactions for optimal transition state interactions while providing a route for the ingress and egress of substrates and products, rather than generic rate enhancements from solvent exclusion (*Wolfenden, 1974, 1976*; *Alber et al., 1987*; *Herschlag, 1988*; *Pompliano et al., 1990*; *Herschlag, 1991*).

## Closing remarks

A deeper investigation into the energetics and energetic interconnections within an active site—that of *E. coli* AP—has led to new knowledge, new models for catalytic effects, and a quantitative assessment of the evolutionary probability of establishing an active site network. Nevertheless, this is just one study, and it remains to be determined how similar or different the behaviors of active sites of other enzymes are. In addition, while extending beyond traditional studies, our work investigates a miniscule fraction of the possible connections and interrelationships in AP. Of particular interest will be understanding the interplay between the broader scaffold and the active site residues along with their functional roles and energetic connections, and such studies will ultimately rely on the development of methods that are high-throughput and highly quantitative.

## Materials and methods

### Protein expression and purification

WT and mutant AP were purified from a fusion construct containing an N-terminal maltose binding protein (MBP) tag and a C-terminal strepII tag with a factor Xa cleavage site between it and the natural C-terminal end of AP. D101 was mutated to alanine, R166 to serine, D153 to alanine, K328 to alanine, and the $Mg^{2+}$ ion ligand E322 to tyrosine to prevent the $Mg^{2+}$ from binding in the active site; these mutations were made alone and in combination. To test for idiosyncratic effects, mutations to alternative residues were tested (*Appendix 1 Table 1*).

*E. coli* SM547(DE3) cells were transformed with the MBP-AP-strepII constructs and were grown to an $OD_{600}$ of 0.6 in rich media and glucose (10 g of tryptone, 5 g of yeast extract, 5 g of NaCl, and 2 g of glucose per liter) with 50 µg/ml of carbenicillin at 37°C. IPTG was added to a final concentration of 0.3 mM to induce protein expression. Cultures were then grown at 30°C for 16–20 hr.

Cells were harvested from 2 l culture by centrifugation at 4400×g for 20 min and lysed with osmotic shock. The cell pellet was resuspended in 800 ml 20% sucrose solution (30 mM Tris-HCl, pH 8.0, 1 mM EDTA) and incubated at room temperature for 10 min on a shaking table. The cells were pelleted by centrifugation at 13,000×g for 10 min. The pelleted cells were resuspended in 800 ml ice cold water at 4°C. The cells were incubated on a shaking table at 4°C for 10 min and then pelleted at 13,000×g for 20 min. The supernatant was adjusted to 10 mM Tris-HCl, pH 7.4, 200 mM NaCl, and 10 μM ZnCl$_2$. The sample was then passed over a 10 ml amylose resin (New England BioLabs, Ipswich, MA) gravity column. All mutants were purified with fresh amylose resin to prevent inadvertent co-purification with other mutants. The amylose column was washed with 10 column volumes of 10 mM Tris-HCl, pH 7.4, 200 mM NaCl, and 10 μM ZnCl$_2$ and eluted with the same buffer supplemented with 10 mM maltose. Protein-containing fractions were concentrated by centrifugation through a 10 kDa cutoff filter (Amicon) and buffer exchanged at least twice into 10 mM sodium MOPS, pH 7.0, 50 mM NaCl, 100 μM ZnCl$_2$, 1.0 mM MgCl$_2$ unless the E322Y mutation was present, in which case MgCl$_2$ was omitted.

For all enzymes, purity was determined to be >95% by SDS-PAGE gel electrophoresis based on staining with Coomassie Blue. To further test for a possible contaminant and to determine the reproducibility of the results, nine out of the 28 mutants were re-expressed and re-characterized: K328A, R166S, E322Y, D101A/D153A, D101A/E322Y, R166S/K328A, D101A/D153A/R166S, D153A/R166S/E322Y/K328A, and D101A/D153A/R166S/E322Y/K328A. Independent preparations gave activities within twofold of one another. To most strongly test whether the observed activities might arise from an enzyme contaminant, the construct with all five residues mutated was further mutated by removal of the serine nucleophile (to give S102G/D101A/D153A/R166S/E322Y/K328A AP). This mutant had no measurable activity above background and a reaction rate at least 100-fold lower than that for any of the AP mutants reported herein, suggesting that the observed activities do not arise from a contaminant.

To control for potential unintended complications specific to the mutant residue introduced, several additional mutations were tested (*Appendix 1 Table 1*). The values of $k_{cat}/K_M$ with mutations to different residues were within twofold in all cases. Alternative AP variants investigated from previous papers are also added to the table and have similar efficiencies.

## Kinetic assays

Activity measurements were performed in 0.1 M MOPS, pH 8.0, 0.5 M NaCl, 100 μM ZnCl$_2$, and 500 μM MgCl$_2$ at 25°C in a UV/Vis Lambda 25 spectrophotometer (Perkin Elmer, Waltham, MA), unless otherwise noted; for mutants containing E322Y, MgCl$_2$ was excluded. The formation of free *p*-nitrophenolate from hydrolysis of the substrate *p*-nitrophenol phosphate (pNPP) was monitored continuously at 400 nm.

Rate constants were determined from initial rates, and the activity of the free enzyme, $k_{cat}/K_M$, was determined. (Rate measurements were limited to $k_{cat}/K_M$ measurements because $k_{cat}$ for WT represents dissociation of product rather than a chemical step and because the rate-limiting step for $k_{cat}$ could vary with mutation. We ensured that the chemical step was rate limiting for the $k_{cat}/K_M$ comparisons carried out herein, as described in *Table 3* and described in Appendix 1.)

At least two different enzyme concentrations and at least seven different substrate concentrations were used for each enzyme. Enzyme concentrations were varied by at least fivefold, and substrate concentrations were extended to at least fivefold below the $K_M$ value for each enzyme based on $K_M$ values determined over wider ranges of substrate. Reaction rates were linear in enzyme concentration at each substrate concentration for each enzyme, and no reaction was observed without added enzyme. Values of $k_{cat}/K_M$ determined from linear fits to the lowest substrate concentrations were the same, within error, as values determined from full Michaelis–Menten fits, and R$^2$ values were >0.98 in all cases. For E322Y AP and D153A/E322Y, because of their very low $K_M$ (~0.5 μM), $k_{cat}/K_M$ was determined from rate measurements in the presence of inhibitory P$_i$ and the independently measured inhibition constant of P$_i$ using an alternative, high $K_M$ substrate, as described previously (*Zalatan et al., 2008*).

Errors were estimated from two independent kinetic measurements, and comparisons with independent enzyme preparations for nine of the AP variants gave the same values and similar error estimates as the same preparation used on separate days.

All mutants were incubated at room temperature for at least a week to test for Zn$^{2+}$ activation. The E322Y mutants exhibited time-dependent Zn$^{2+}$ activation, as observed previously, and were shown to level to a maximal rate (*Zalatan et al., 2008*). Mutants with E322 not mutated were incubated in

**Table 3.** Kinetic constants for Me-P hydrolysis

| AP mutant | $k_{cat}/K_M$ (M$^{-1}$s$^{-1}$) | | |
| --- | --- | --- | --- |
| | pNPP | Me-P | $\frac{(k_{cat}/K_M)_{pNPP}}{(k_{cat}/K_M)_{Me-Pi}}$ |
| WT*,† | $3.3 \times 10^7$ | $1.2 \times 10^6$ | 28 |
| | | $3.5 \times 10^5$ | 80 |
| R166S† | $1.0 \times 10^5$ | 110 | 910 |
| | | 56 | 1800 |
| E322Y‡ | $7.2 \times 10^3$ | 1.6 | 4500 |
| D101A | $9.9 \times 10^6$ | $2.7 \times 10^3$ | 3600 |
| D153A | $2.8 \times 10^6$ | $1.1 \times 10^3$ | 2500 |
| D101A/D153A | $3.3 \times 10^5$ | 61 | 5400 |

*$(k_{cat}/K_M)_{obsd}$; the chemical step is not rate limiting.
†Values of $k_{cat}/K_M$ of $1.2 \times 10^6$ M$^{-1}$s$^{-1}$ and 110 M$^{-1}$s$^{-1}$ for WT and R166S, respectively for Me-P was obtained previously from reference (**O'Brien and Herschlag, 2002**; **O'Brien et al., 2008**). This difference in value would not affect the conclusions herein.
‡From reference (**Zalatan et al., 2008**).
The efficiency of Me-P hydrolysis was measured for mutants with pNPP activities close to the rate of diffusion. The ratios measured for the enzyme with pNPP and Me-P were close to what had been measured previously for R166S and E322Y, two mutants for which chemistry is rate limiting for both substrates. The ratios suggest that chemistry is rate limiting for the pNPP hydrolysis in D101A and D153A.

storage buffer with Mg$^{2+}$ added (10 mM sodium MOPS, pH 7.0, 50 mM NaCl, 100 μM ZnCl$_2$, 1 mM MgCl$_2$). To test for kinetic effects arising from decreased metal affinity of the mutants compared to WT, mutants R166S, D101A/R166S/D153A/K328A, D101A/D153A, E322Y, D101A/R166S/D153A/E322Y/K328A, and D101A/R166S were also incubated with varying Zn$^{2+}$ and Mg$^{2+}$ concentrations for a week. The following metal ion concentrations were used in the incubation experiment: 100 μM ZnCl$_2$; 1.0 mM ZnCl$_2$; 100 μM ZnCl$_2$ and 100 μM MgCl$_2$; 100 μM ZnCl$_2$ and 1.0 mM MgCl$_2$; 100 μM ZnCl$_2$ and 10 mM MgCl$_2$; 1.0 mM ZnCl$_2$ and 10 mM MgCl$_2$. Only AP mutants with E322Y exhibited time-dependent activation, as observed previously (**Zalatan et al., 2008**); the Mg$^{2+}$ concentration did not affect the Zn$^{2+}$ activation; and the activities of the E322Y mutants were not dependent on the presence of Mg$^{2+}$. The activities of mutants R166S, D101A/R166S/D153A/K328A, D101A/D153A, and D101A/R166S do not increase with higher concentrations of Mg$^{2+}$.

Phosphate monoester hydrolysis by D101A, D153A, R166S, and D101A/D153A AP was also measured with methyl phosphate (Me-P), using the same reaction buffer and conditions as for pNPP. The formation of the inorganic phosphate (P$_i$) product from Me-P hydrolysis was monitored discontinuously by withdrawing aliquots from ongoing reactions, quenching in 6 M guanidine-HCl, and detecting P$_i$ with a modified Malachite Green assay (**Lanzetta et al., 1979**) at eight or more specified times. Rate constants were determined from initial rates, and activity of the free enzyme, $k_{cat}/K_M$, was determined. At least two different enzyme concentrations and at least seven different substrate concentrations were used for each enzyme. Enzyme concentrations were varied by at least fivefold, and substrate concentrations were extended to at least fivefold below the $K_M$ value for each enzyme based on $K_M$ values determined over wider ranges of substrate. Reaction rates were linear in enzyme concentration at the lowest substrate concentration for each enzyme, and no reaction was observed without added enzyme. R$^2$ values were >0.98 in all cases.

## Test of Mg$^{2+}$ occupancy

To test if the most mutated AP mutant, AP minimal (D101A/R166S/D153A/E322Y/K328A), had Mg$^{2+}$ in the active site and full Zn$^{2+}$ occupancy, we carried out atomic emission spectroscopy, as previously used for the E322Y single mutant, with the AP minimal mutant (**Zalatan et al., 2008**). The metal ion occupancies were consistent with an active site saturated with Zn$^{2+}$ and lacking Mg$^{2+}$ (Zn$^{2+}$:protein ratio 2.49; Mg$^{2+}$: protein ratio 0.01; P$_i$: protein ratio 0.06). As described above, kinetic experiments were also carried out to test for Mg$^{2+}$ activation.

## Crystallization and structure determination

The MBP tag used for purifying D101A/D153A was cleaved with factor Xa, and the enzyme was separated from the tag over a 5 ml HiTrap Q HP column (GE Healthcare, Amersham, UK). The purified enzyme was buffer exchanged into 10 mM sodium Tris, pH 7.0, 50 mM NaCl, and 100 μM ZnCl$_2$ and concentrated to 5.1 mg/ml. Equal volumes of enzyme and precipitant solution (22% PEG3350, 0.1 mM Bis-Tris, pH 5.0, 0.2 mM ammonium sulfate) were mixed and placed over a reservoir of 1 ml precipitant solution to crystalize by the hanging drop method. No inorganic phosphate (P$_i$) was added to the precipitant solution, but 0.8 mM contaminating P$_i$ was found in the

crystallization solution using a Malachite Green assay (*Lanzetta et al., 1979*). Crystals were soaked in a cryoprotectant solution of 30% glycerol, 0.1 mM Bis-Tris, pH 5.0, and 0.2 mM ammonium sulfate prior to being frozen in liquid nitrogen. Crystallographic data were collected at the Stanford Linear Accelerator at beamline 11-1.

The D101A/D153A mutant of AP crystallized in space group $P6_322$ with one dimer per asymmetric unit. Data were integrated with MOSFLM (*Battye et al., 2011*) and scaled and merged with AIMLESS (*Evans and Murshudov, 2013*). Five percent of reflections were set aside for calculation of $R_{free}$. Molecular replacement was performed with PHASER (*McCoy et al., 2007*) using WT AP (PDB 3TG0) stripped of phosphate and metal ions as a search model. Rounds of alternating manual and automated refinement were performed with COOT and REFMAC5, respectively (*Emsley et al., 2010*; *Murshudov et al., 2011*). Stereochemistry was assessed with MOLPROBITY, and images were generated with PYMOL (*Schrödinger, 2010*). The PDB deposition ID is 4YR1.

## Acknowledgements

This work was funded by a grant from the US National Institutes of Health to DH (GM64798). The SSRL Structural Molecular Biology Program is supported by the Department of Energy, Office of Biological and Environmental Research, and by the National Institutes of Health, National Center for Research Resources, Biomedical Technology Program, and the National Institute of General Medical Sciences. We thank SSRL scientist Clyde Smith for excellent support. AP was funded in part by a National Science Foundation Graduate Research Fellowship. SR was funded by the Max-Planck Society Germany. We thank James Fraser (UCSF) and Artem Lyubimov (Stanford) for guidance and review on refinements of diffraction data and members of the Herschlag laboratory for discussions and comments on the manuscript. We thank Celerino Abad-Zapatero and Vincent Stoll for information about their published D101S AP structure.

## Additional information

### Funding

| Funder | Grant reference | Author |
|---|---|---|
| National Institutes of Health (NIH) | GM64798 | Daniel Herschlag |
| National Institutes of Health (NIH) | GM049243 | Daniel Herschlag |
| U.S. Department of Energy | Office of Biological and Environmental Research | Fanny Sunden |
| National Institutes of Health (NIH) | | Fanny Sunden |
| National Center for Research Resources (NCRR) | | Fanny Sunden |
| National Institute of General Medical Sciences (NIGMS) | | Fanny Sunden |
| Biomedical Technology Program | | Fanny Sunden |
| National Science Foundation (NSF) | Graduate Research Fellowship | Ariana Peck |
| Max-Planck-Gesellschaft | | Susanne Ressl |

The funders had no role in study design, data collection and interpretation, or the decision to submit the work for publication.

### Author contributions

FS, Conception and design, Acquisition of data, Analysis and interpretation of data, Drafting or revising the article; AP, DH, Conception and design, Analysis and interpretation of data, Drafting or revising the article; JS, Analysis and interpretation of data, Drafting or revising the article; SR, Acquisition of data, Analysis and interpretation of data, Drafting or revising the article

# Additional files

## Major datasets

The following dataset was generated:

| Author(s) | Year | Dataset title | Dataset ID and/or URL | Database, license, and accessibility information |
|---|---|---|---|---|
| Peck A, Herschlag D | 2015 | Crystal Structure of E. Coli Alkaline Phosphatase D101A/D153A in complex with inorganic phosphate | http://www.rcsb.org/pdb/ search/structidSearch.do? structureId=4YR1 | Publicly available at RCSB Protein Database (4YR1). |

The following previously published datasets were used:

| Author(s) | Year | Dataset title | Dataset ID and/or URL | Database, license, and accessibility information |
|---|---|---|---|---|
| Dealwis CG, Brennan C, Christianson K, Mandecki W, Abad-Zapatero C | 1995 | Three-dimensional structure of the D153F mutant of E. Coli alkaline phosphatase: a mutant with weaker magnesium binding and increased catalytic activity | http://www.rcsb.org/pdb/ explore/explore.do? structureId=1AJC | Publicly available at RCSB Protein Database (1AJC). |
| Dealwis CG, Brennan C, Christianson K, Mandecki W, Abad-Zapatero C | 1995 | Three-dimensional structure of the D153F mutant of E. Coli alkaline phosphatase: a mutant with weaker magnesium binding and increased catalytic activity | http://www.rcsb.org/pdb/ explore/explore.do? structureId=1AJD | Publicly available at RCSB Protein Database (1AJD). |
| Stec B, Holtz KM, Kantrowitz ER | 2000 | Structure of E. Coli alkaline phosphatase without the inorganic phosphate at 1.75A resolution | http://www.rcsb.org/pdb/ explore/explore.do? structureId=1ED9 | Publicly available at RCSB Protein Database (1ED9). |
| Bobyr E, Lassila JK, Wiersma-Koch HI, Fenn TD, Lee JJ, Nikolic-Hughes I, Hodgson KO, Rees DC, Hedman B, Herschlag D | 2012 | E. coli alkaline phosphatase with bound inorganic phosphate | http://www.rcsb.org/pdb/ explore/explore.do? structureId=3TG0 | Publicly available at RCSB Protein Database (3TG0). |
| Zalatan JG, Fenn TD, Herschlag D | 2008 | Structure of E322Y Alkaline Phosphatase in Complex with Inorganic Phosphate | http://www.rcsb.org/pdb/ explore/explore.do? structureId=3DYC | Publicly available at RCSB Protein Database (3DYC). |

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

## Appendix 1

## Additional expressed and characterized AP mutants.

Several alternative mutations to those displayed in **Table 1** and **Figure 2** have been made. The catalytic efficiency of the mutational variant used in the study is highlighted in bold for easy reference, and additional mutants are listed in **Appendix 1 Table 1**. Relevant mutants tested in previous studies are also listed.

**Appendix 1 Table 1**. Alternative mutations

| AP mutant | $k_{cat}/K_M$ (M$^{-1}$s$^{-1}$) | $k_{rel}$ |
|---|---|---|
| D101A | **9.9 (2.0) × 10$^6$** | **(1)** |
| D101A* | 1.0 × 10$^7$ | 1.0 |
| D101G | 7.6 (1.3) × 10$^6$ | 1.5 |
| D101S | 9.1 (1.3) × 10$^6$ | 1.1 |
| D101S† | 3.1 × 10$^6$ | 3.2 |
| D101S‡ | 1.7 × 10$^6$ | 5.8 |
| D153A | **2.8 (0.4) × 10$^6$** | **(1)** |
| D153A§ | 1.5 × 10$^6$ | 1.9 |
| D153G‡ | 1.6 × 10$^6$ | 1.8 |
| R166S# | **1.0 × 10$^5$** | **(1)** |
| R166S¶ | 5.8 × 10$^4$ | 1.7 |
| R166A# | 2.0 × 10$^4$ | 5.0 |
| E332Y** | **7.2 (2.2) × 10$^3$** | **(1)** |
| E322A** | 8.9 × 10$^3$ | 0.8 |
| K328A** | **7.5 (2.4) × 10$^5$** | **(1)** |
| K328A†† | 3.6 × 10$^6$ | 0.2 |
| R101A/R166S | **5.8 (0.2) × 10$^4$** | **(1)** |
| R101G/R166S | 2.7 (0.3) × 10$^4$ | 2.1 |
| R166S/D153A/E322Y/K328A | **4.4 (0.1)** | **(1)** |
| R166S/D153A/E322A/K328A | 2.7 | 1.6 |
| D101A/R166S/D153A/E322Y/K328A | **1.7 (0.3) × 10$^{-1}$** | **(1)** |
| D101G/R166S/D153A/E322Y/K328A | 1.3 (0.2) × 10$^{-1}$ | 1.3 |
| D101N/R166S/D153A/E322Y/K328A | 1.9 (0.3) × 10$^{-1}$ | 0.9 |

*Reference (**Herschlag, 1988**).
†Reference (**Freedman et al., 2009**).
‡Reference (**Herschlag, 1991**).
§Reference (**Wolfenden, 1976**).
#Reference (**Eisenmesser et al., 2002**).
¶Reference (**Lanzetta et al., 1979**).
**Reference (**Halabi et al., 2009**).
††Reference (**Battye et al., 2011**).

**Appendix 1 Table 2**. Comparison of predicted and observed catalytic efficiencies of the 32 possible AP mutants

| AP mutant | $(k_{cat}/K_M)^{Predicted}$ (M$^{-1}$s$^{-1}$) | Ratio measured/predicted |
|---|---|---|
| WT | 4.2 × 10$^8$ | 1.5 |
| D101A | 9.7 × 10$^6$ | 1.0 |

*Appendix 1 Table 2. Continued on next page*

*Appendix 1 Table 2. Continued*

| AP mutant | $(k_{cat}/K_M)^{Predicted}$ $(M^{-1}s^{-1})$ | Ratio measured/predicted |
|---|---|---|
| R166S | $8.5 \times 10^4$ | 1.2 |
| D153A | $1.9 \times 10^6$ | 1.5 |
| E332Y | $6.6 \times 10^3$ | 1.1 |
| K328A | $1.2 \times 10^6$ | 0.6 |
| D101A/R166S | $9.6 \times 10^4$ | 0.6 |
| D101A/D153A | $3.9 \times 10^5$ | 0.8 |
| D101A/E322Y | 4.2 | 0.7 |
| D101A/K328A* | $2.8 \times 10^4$ | N/A |
| R166S/D153A | $1.9 \times 10^4$ | 0.7 |
| R166S/E322Y | 1.3 | 1.2 |
| R166S/K328A | $2.5 \times 10^2$ | 1.0 |
| D153A/E322Y | $2.2 \times 10^3$ | 1.0 |
| D153A/K328A | $4.1 \times 10^5$ | 1.1 |
| E322Y/K328A | $1.4 \times 10^3$ | 1.1 |
| D101A/R166S/D153A | $2.1 \times 10^4$ | 0.7 |
| D101A/R166S/E322Y* | $4.2 \times 10^{-2}$ | N/A |
| D101A/R166S/K328A* | $2.8 \times 10^2$ | N/A |
| D101A/D153A/E322Y* | 13 | N/A |
| D101A/D153A/K328A | $8.5 \times 10^4$ | 1.4 |
| D101A/E322Y/K328A | 0.92 | 1.2 |
| R166S/D153A/E322Y | 22 | 0.9 |
| R166S/D153A/K328A | $4.0 \times 10^3$ | 0.9 |
| R166S/E322Y/K328A | 0.29 | 1.3 |
| D153A/E322Y/K328A | $4.8 \times 10^2$ | 0.7 |
| D101A/R166S/D153A/E322Y | 0.68 | 1.4 |
| D101A/R166S/D153A/K328A | $4.5 \times 10^3$ | 1.2 |
| D101A/R166S/E322Y/K328A | $9.0 \times 10^{-3}$ | 2.2† |
| D101/D153A/E322Y/K328A | 2.8 | 0.7 |
| R166S/D153A/E322Y/K328A | 4.8 | 0.9 |
| D101A/R166S/D153A/E322Y/K328A | 0.15 | 1.1 |

*Obtained from the predicted values in this table and the observed values in **Table 1**.
†Only an upper limit for this mutant could be measured.

## A unifying quantitative model for the energetic behavior of the five-residue AP active site network.

Removal of individual residues from WT AP gives different catalytic effects than introduction of the same residues in a minimal background lacking all five of the investigated residues (**Figure 3A,B**). These differences indicate that an independent model to account for the effect of the individual residues cannot accurately reproduce the data. In a mathematical sense, more than five variables are needed. Here we develop a minimal mathematical model to account for all 32 AP variants. This model accurately reproduces the 28 measured rate constants (within twofold; table below) and provides predictions for the four unmeasured rate constants.

Our analyses in the 'Results and discussion' sections identify three functional units. If these units fully account for the functional interactions between side chains then we should be able to

accurately fit the data with a mathematical model with the following components (*Equation 2* below): 1. Five baseline values for addition of each residue individually (first bracketed terms in *Equation 2*); and 2. One term for each of the functional units (three total), each with a different functional form to describe the energetic behavior of that functional unit—that is, independent energetic effects in the D101/R166/D153 unit (second bracketed term); cooperative effects in the D153/Mg$^{2+}$/K328 unit (third bracketed term); and anti-cooperative effects in the D101/Mg$^{2+}$ unit (fourth bracketed term). The superscript for the residue in question has a value of 1 if the WT residue is present, or 0 if the mutant residue is present. The numerical values used in *Equation 2* were obtained by optimization via a least squares fitting algorithm run in MatLab.

$$v_{mutant} = v_{minimal} \times \left\{ 0.89^{D101} \times 19^{R166} \times 0.061^{D153} \times 850^{Mg^{2+}} \times 4.6^{K328} \right\}$$
$$\times \left\{ 9.0^{((D101 \times D153) \times R166)} \times 5.4^{((D101+D153) \times R166)} \right\} \times \left\{ 75^{(D153 \times K328 \times Mg^{2+})} \right\}$$
$$\times \left\{ 36^{(D101+Mg^{2+}-D101 \times Mg^{2+})} \right\}.$$

## Tests of whether the chemical step is rate limiting and estimation of the free energy barrier for the chemical step for cases when it is not rate limiting.

The chemical step of pNPP hydrolysis by WT AP has previously been shown to not be rate limiting (subsaturating conditions: $k_{cat}/K_M$; [*Pompliano et al., 1990*]), whereas the hydrolysis rate of alkyl phosphate esters such as methyl phosphate (Me-P) is limited by the chemical step. For AP mutants with $k_{cat}/K_M$ values for pNPP hydrolysis that are substantially smaller than that for WT AP and therefore likely are rate-limited by the chemical step instead of substrate association, the ratio of reactivity of pNPP to Me-P (or other alkyl phosphates) is in the range of 1000–5000 (*Pompliano et al., 1990*; *Eisenmesser et al., 2002*). (The lower $k_{cat}/K_M$ value for alkyl phosphates presumably arises in large part due to their lower intrinsic reactivity, which is reflected in their high leaving group p$K_a$ relative to pNPP p$K_a$ = ~16 and 7 for alkyl alcohols and p-nitrophenol, respectively [*Evans and Murshudov, 2013*].) We therefore determined $k_{cat}/K_M$ values for Me-P hydrolysis for several of the faster reacting mutants used in this study in addition to the $k_{cat}/K_M$ values for pNPP, and we calculated the ratio of pNPP to Me-P reactivity (*Table 3*) (*Eisenmesser et al., 2005*). The similar values of these ratios for all AP variants other than WT strongly suggest that the chemical step is rate limiting or nearly rate limiting in all cases. The small variations of the ratios could represent experimental error, small idiosyncrasies of individual mutants, or a not-quite fully rate-limiting chemical step for the mutants with the smaller ratios (*Table 3*). These small differences, if present in the chemical step, would not affect any of the conclusions drawn herein. In principle we could have used Me-P for all of the comparisons instead of pNPP and a correction for WT (see below), but the faster intrinsic reactivity of pNPP has allowed us to measure AP mutants that are over a billion-fold slower than WT and will allow future comparisons with considerably less reactive AP substrates (*McCoy et al., 2007*; *Emsley et al., 2010*).

As alluded to above, to provide a $k_{cat}/K_M$ value for WT AP that could be compared with the mutant APs, that is, that represented the same rate-limiting step, we calculated the $k_{cat}/K_M$ value expected for pNPP if the chemical step were rate limiting, that is, if the binding equilibration were sufficiently fast such that association were not rate limiting. To accomplish this we used the procedure of O'Brien et al. (*Eisenmesser et al., 2002*), assuming that the ratio of the barrier height for the chemical step for Me-P vs pNPP is the same for WT AP as it is for R166S AP, a reasonable assumption given the similarity of this ratio for a range of mutants and the observation that R166 interacts with the non-bridging phosphoryl oxygen atoms that are common in the two substrates and does not interact with the leaving groups, which are different. The ratio of $k_{cat}/K_M$ values for Me-P hydrolysis and pNPP hydrolysis for R166S

$(1.0 \times 10^5$ M$^{-1}$s$^{-1}$/57 M$^{-1}$s$^{-1}$ = 1800) was multiplied with the Me-P activity measured for WT $(3.5 \times 10^5$ M$^{-1}$s$^{-1}$) to give the estimated $k_{cat}/K_M$ value for pNPP hydrolysis by WT AP of $6.3 \times 10^8$ M$^{-1}$s$^{-1}$ (**Eisenmesser et al., 2002**, **2005**).

## Appendix 2

# Hydrogen bond distances and angles between R166 and P$_i$.

*Appendix 2 Table 1* below lists the bond lengths and angles between the closest atoms of the R166 side chain and bound P$_i$ for WT (PDB 3TG0) and D101A/D153A AP. The density corresponding to the ligand in D101A/D153A AP was best fit by two partially occupied P$_i$ ions in positions distinct from the position in WT AP. Due to the spread of the electron density from the P$_i$ ion and the lower resolution of the D101A/D153A model (2.5 Å compared to 1.2 Å for WT AP), the reported distances and angles for this structure are subject to greater coordinate error than those measured for WT AP.

**Appendix 2 Table 1**.

| AP variant | Donor–Acceptor‡ | Distance* (Å) | Angle† (°) |
|---|---|---|---|
| WT | NH1-O1 | 2.8 | 170 |
| | NH2-O2 | 2.8 | 177 |
| D101A/D153A | NH1-O1 (1) | 3.0 | 158 |
| | NH2-O1 (1) | 3.4 | 139 |
| | NH1-O2 (1) | 4.9 | 131 |
| | NH2-O2 (1) | 4.3 | 142 |

*Distances between donor and acceptor atoms.

†The angle between the donor, predicted hydrogen, and acceptor atoms.

‡NH1 and NH2 refer to the nitrogen atoms of the guanidinium group of R166. O1 and O2 refer to the relevant oxygen atom of the bound P$_i$; in the case of D101A/D153A, the number in parenthesis refers to the partially occupied P$_i$ labeled in the *Appendix 2 Figure 1* below. Only distances and angles to one of these phosphate ions in the monomer shown in *Appendix 2 Figure 1* panel (B) are listed in the table; the other P$_i$ sterically clashes with the active-site facing rotamer of R166 and thus presumably is populated only when the R166 rotamer is flipped away from the active site. Distances and angles are not reported for the other subunit (shown in *Appendix 2 Figure 1C* below) because in this active site, R166 is flipped out of the active site so that it is not within hydrogen bonding distance of the bound P$_i$. For reference, the WT active site is shown in *Appendix 2 Figure 1* panel (A) below.

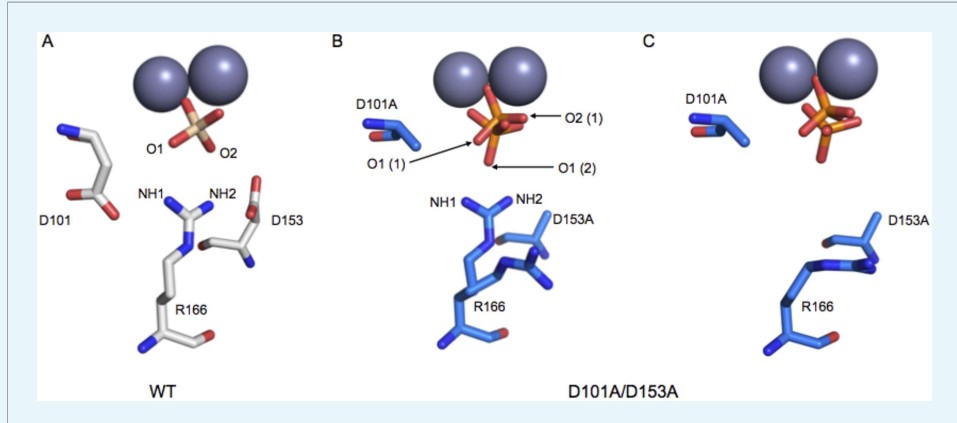

**Appendix 2 Figure 1**. Labeling scheme for describing hydrogen bond distances and angles between R166 and P$_i$.

## Appendix 3

### Evolutionary landscape probability and mean waiting time calculations.

In deriving the mean waiting times for evolutionary models, the following simplifications were made: population-based factors such as differential population sizes based on selective advantages were not included, the probability for mutating to any of the 20 amino acids has been assumed to be the same, and it was assumed that only the WT residue can confer a selective advantage. A mean waiting time ($t_{mean}$) is the same as a reaction halftime for processes that can be described by an exponential: that is, $t_{mean} = (\ln 2/k) = 0.693/k$. In the main text, we only provide mean waiting times in cases that can be described by an exponential form, as simulations are required for models with non-exponential forms and the conclusions in this study do not rest on such details.

A hidden Markov Model can also be used to calculate the mean waiting time. Assume that we allow five positions to mutate in any order to their WT residues using a discrete time process, and when any position is mutated to the WT residue, it is fixed. We additionally assume that at each step, mutations are modeled as occurring uniformly at random at any position that has not yet been fixed to the WT residue. (For simplicity, we consider mutations at the amino acid rather than the nucleotide level; this simplification does not significantly alter the conclusions.) We also assume that two or more residues cannot mutate to their corresponding WT residues in the same time step. Define $I_j$, $1 \leq j \leq 5$ as the $j$th position that acquires its WT residue (i.e., $I_1 = 3$ if the third position is mutated to WT before all others). Let $p_j^*$ for $1 \leq j \leq 5$ be defined as follows: at time $t$, if $j$ positions have been mutated to their WT residue, $p_j^*$ is the probability that no other positions will be mutated to their WT residue at the next time step. Then, it follows that:

$$p_j^* = \left(\frac{19}{20}\right)^{(5-i)}. \tag{3}$$

If $T$ is the total waiting time for all positions to acquire their WT residues, and $E(T_{Ij})$ is the mean waiting time to achieve optimality at residue $I_j$,

$$E(T) = E(T_{I_1}) + \ldots + E(T_{I_5}) = \sum_{i=1}^{5} \frac{1}{1 - (19/20)^i} = \sim 47, \tag{4}$$

by the application of the simple formula for expectation (**Equation 3**), and a mean waiting time of 47 is calculated.

$$E(T_{I_j}) = \sum_{k=0}^{\infty} P(T_{I_j} > k) = \sum_{k=0}^{\infty} \left(p_j^*\right)^k = \frac{1}{1 - p_j^*}. \tag{5}$$

For the fully independent model, we model the mutational process as a fully connected random walk on all sequences where each position is mutated to the WT residue uniformly at random in each time step. In this case the mean waiting time can be determined according to **Equation 6** and is calculated to be $3.2 \times 10^6$.

$$E(T) = \sum_{k=0}^{\infty} P(T > k) = \sum_{k=0}^{\infty} \left(1 - \left(\frac{1}{20}\right)^5\right)^k = \frac{1}{\left(\frac{1}{20}\right)^5}. \tag{6}$$

