## [Decision Letter]

Thank you for sending your work entitled “Deep Site-directed Mutagenesis Reveals Interconnected Functional Units in the *E. coli* Alkaline Phosphatase Active Site” for consideration at *eLife*. Your article has been favorably evaluated by John Kuriyan (Senior editor and Reviewing editor) and three reviewers.

The Reviewing editor and the reviewers discussed their comments before we reached this decision, and the Reviewing editor has assembled the following comments (see below) to help you prepare a revised submission.

The authors of this paper have shown ingenuity in addressing a challenging question: the levels of organization of an enzyme on which selective pressure is brought to bear. Building on this group's earlier unraveling of the mechanism of action of alkaline phosphatase, they explore the activities of various catalytic groups in various mutant backgrounds. The detailed findings and their interpretation will interest the specialist, but the larger picture will interest a broader readership. Not surprisingly, the results support an evolutionary strategy of “divide and conquer” that involves functional units, or overlapping groups of amino acid residues, but avoids requiring that extremely high levels of cooperativity spring into existence.

The authors perform a thorough investigation of a network of five catalytic residues in *E. coli* Alkaline Phosphatase (AP). Given the structural characterization and amenability of the system to extensive mutagenesis and kinetic analysis, AP is a perfect model system for this study. Here, the authors relied on properties of two AP-derived model systems: a WT enzyme that contains individual point mutations of five catalytically critical residues: D101, D153, R166, E322 (a mutation that abrogates binding of catalytic Mg^2+^) and K328 and a minimal enzyme, where all five of these residues are mutagenized and individual WT residues added back into this minimal background. Measurements of *k*_*cat*_*/K*_M_ in these mutants have allowed authors to determine if the catalytic effects of any of these residues individually are dependent on the context of any of the other four.

The study reveals three functional subunits, which while structurally interconnected are coupled in terms of free energy. The conclusions are based on elegantly designed, very clean experiments where the side chain of interest is added into the WT or the minimal background, and effect of *k*_*cat*_*/K*_M_ measured. While, for example, the catalytic effectiveness of R166 is enhanced by D101 and D153, it is unaffected by the presence or absence of K328 and Mg^2+^. These experiments led to a conclusion that D101/R166/D153 form a functional unit, which is supported by the observed distortion and flipping away of the guanidine group in X-ray structures of previously published D153G and newly acquired D101A/D153A AP, while the positioning of Arg in the structure of E322Y AP is the same as in the WT. Similarly, D153/ Mg^2+^/K328 form a functional unit, and so do D101/Mg^2+^. While all three functional subunits are uncoupled in terms of free energy, they are interconnected as they share residues in common. The authors speculate that such arrangement is preferred, given the limited space in the active site. Additionally, they postulate that multiple uses of a same residue is, evolutionarily, a more feasible scenario.

Overall, the study is highly significant and it provides a blueprint for interrogation of functional interconnectivity in an active site.

Major issue to address:

The manuscript, as presently written, fails to do justice to the large body of extant work on internal coupling in proteins. The article begins with the statement that the establishment of functional connected catalytic units within an enzyme's active site has not been previously accomplished. That may be true for alkaline phosphatase but as a general statement it is false. Studies on triose phosphate isomerase, ribonuclease, and dihydrofolate reductase have established the presence of a coupled network of catalytic residues both within and outside the catalytic site that would qualify under the author's definition as interconnected catalytic units. Modern computational methods when combined with experimental data provide for powerful insights because the entire enzyme can be taken into consideration. Small individual units are often found embedded in larger secondary protein elements with the latter being the “unit”. Finally, the conclusion that the evaluation of the five residues subject to mutation arrived evolutionarily in a stepwise rather than a cooperative process is widely anticipated based on numerous literature articles.

There are too many key papers that have not been cited:

Dihydrofolate reductase:

a) Howell et al., 1993 (suppressor mutations);

b) Wright et al., 2005 (residue connectivity through relaxation NMR);

c) Kohen et al., 2014 (connectivity through temperature dependent isotope efforts);

d) Benkovic et al., 2002 (double/triple mutation of active site residues); and

e) Romanathan/Hammes-Schiffer (molecular mechanics calculations).

Triose phosphate isomerase:

Knowles et al., 1990, and Petsko et al., 1987.

Ribonuclease:

Numerous authors.

Other specific issues to address in the manuscript:

1) The adjective “deep” should be removed from the Title and a section heading in the paper. Given the common understanding of the term “Deep Sequencing”, the use of “deep” in the context of this paper is confusing, since it implies much more extensive mutagenesis than is actually done.

2) The assumption of irreversibility in the calculation of the evolution probabilities is for the convenience of the author's calculations. Evolution is stochastic and reversible.

3) The unexpected anti-cooperativity between D101 and Mg^2+^ is particularly interesting. The authors speculate that the functional link is mediated by one of the Zn ions in the active site as well as the Ser102, a nucleophile in the hydrolysis reaction and a residue directly adjacent to D101, and that the anti-cooperativity stems from redundant effects in positioning the serine oxyanion. An alternative explanation is that both D101 and Mg^2+^ may function redundantly not to position the serine nucleophile, but rather to enhance electrophilicity of the phosphorus atom (and thus rate of Ser addition) by pulling electron density away through interactions with the phosphoryl oxygen atoms. In this model, D101 would do so through the guanidine group of Arginine, while the effect of Mg^2+^ is water mediated. How does D101(+/-), D153+, R166+, K328 + compare to D101(+/-), D153+, R166-, K328+ in the presence or absence of Mg^2+^ (Figure 8)?

4) In the Abstract it is stated that: “Although fundamental to our understanding of enzyme function, evolution and engineering, the properties of these networks have yet to be deeply and systematically explored”. Given the major point raised above, this should be replaced with a more nuanced statement. Likewise, the example given in the Introduction of the fact that a denatured protein still has its catalytic groups and yet is inactive is trivial, and should be replaced by a more substantial discussion. The following paragraph, which starts with a discussion of X-ray structures, is an opportunity to fill in the reader on the extant work on residue coupling in proteins.

5) Again, given the major point raised above, the first paragraph of the paper goes from citations of historical work to citations of very general works on enzyme mechanism. The historical citations hardly seem relevant here, and it would be better to make the opening more substantial in terms of the proper intellectual context for this work.

---

## [Author Response]

*Major issue to address*:

*The manuscript, as presently written, fails to do justice to the large body of extant work on internal coupling in proteins. The article begins with the statement that the establishment of functional connected catalytic units within an enzyme's active site has not been previously accomplished. That may be true for alkaline phosphatase but as a general statement it is false. Studies on triose phosphate isomerase, ribonuclease, and dihydrofolate reductase have established the presence of a coupled network of catalytic residues both within and outside the catalytic site that would qualify under the author's definition as interconnected catalytic units*.

We fully agree and failed to make the important distinction between our work and a body of elegant and important prior work. The prior results make two general important points that are now described in the manuscript (along with the pertinent references): 1) There are energetic interactions between active site residues (e.g., the effects of mutations in DHFR on the ‘top’ and ‘bottom’ of the substrate binding pocket); and 2) There are many examples of long-range effects and long-range coupling between residues in enzymes (e.g., this, most simply, arises in all cases of allostery). By contrast, by examining the full set of mutant combinations, our study delineates a full set of interaction free energies and is able to define functional units among structurally connected active site residues.

*Modern computational methods when combined with experimental data provide for powerful insights because the entire enzyme can be taken into consideration. Small individual units are often found embedded in larger secondary protein elements with the latter being the “unit”*.

We agree that simulations are powerful and can suggest models of energetic connectivity. Nevertheless, computational results from modeling physical processes are not direct experimental observations. Given this distinction, we have referenced models based on phylogenetic data but not those based on physical modeling.

*Finally, the conclusion that the evaluation of the five residues subject to mutation arrived evolutionarily in a stepwise rather than a cooperative process is widely anticipated based on numerous literature articles*.

While it is common knowledge that evolution progresses in a stepwise manner, we were surprised by the enormous extent of the evolutionary advantage of the underlying AP energetics, relative to a fully cooperative network, and others we have discussed this work with have been similarly surprised.

*There are too many key papers that have not been cited*:

Dihydrofolate reductase:

*a) Howell et al., 1993 (suppressor mutations)*;

*b) Wright et al., 2005 (residue connectivity through relaxation NMR)*;

*c) Kohen et al., 2014 (connectivity through temperature dependent isotope efforts)*;

*d) Benkovic et al., 2002 (double/triple mutation of active site residues)*; *and*

e) Romanathan/Hammes-Schiffer (molecular mechanics calculations).

Triose phosphate isomerase:

Knowles et al., 1990, and Petsko et al., 1987.

Ribonuclease:

*Numerous authors*.

We fully agree, as noted above, and have added these and other references.

*Other specific issues to address in the manuscript*:

*1) The adjective “deep” should be removed from the Title and a section heading in the paper. Given the common understanding of the term “Deep Sequencing”, the use of “deep” in the context of this paper is confusing, since it implies much more extensive mutagenesis than is actually done*.

We have removed “deep” as requested. The new Title is: “Extensive Site-directed Mutagenesis Reveals Interconnected Functional Units in the Alkaline Phosphatase Active Site”.

*2) The assumption of irreversibility in the calculation of the evolution probabilities is for the convenience of the author's calculations. Evolution is stochastic and reversible*.

For the fully cooperative case the mutations were treated as fully reversible, as there is no selective pressure to maintain the wild-type residues, until the last residue is added. When there is a selective advantage, then the probability of going ‘back’ is reduced. The extent of reduction is a result of the selective advantage that is conferred, but many factors contribute in addition to the protein’s activity (e.g., the degree to which this activity is limiting for survival, the degree of increase in population size, the relationship between population size and survival and reproduction) Because of this complexity, we simplified the model and assumed a reversion probability of zero, to render the effects most transparent and easiest to interpret. We refer to these simplifications now in the legend to Figure 10.

*3) The unexpected anti-cooperativity between D101 and Mg*^*2+*^
*is particularly interesting. The authors speculate that the functional link is mediated by one of the Zn ions in the active site as well as the Ser102, a nucleophile in the hydrolysis reaction and a residue directly adjacent to D101, and that the anti-cooperativity stems from redundant effects in positioning the serine oxyanion. An alternative explanation is that both D101 and Mg*^*2+*^
*may function redundantly not to position the serine nucleophile, but rather to enhance electrophilicity of the phosphorus atom (and thus rate of Ser addition) by pulling electron density away through interactions with the phosphoryl oxygen atoms. In this model, D101 would do so through the guanidine group of Arginine, while the effect of Mg*^*2+*^
*is water mediated. How does D101(+/-), D153+, R166+*, *K328+ compare to D101(+/-), D153+, R166-, K328+ in the presence or absence of Mg*^*2+*^
*(*Figure 8*)?*

There is evidence against this model. The D101 effect (with Mg^2+^ absent) that is referred to here is independent of the presence or absence of R166, i.e., this 20-fold effect does not require the presence of R166. There is an additional effect of D101 when R166 is present. We have clarified this in the text.

*4) In the Abstract it is stated that: “Although fundamental to our understanding of enzyme function, evolution and engineering, the properties of these networks have yet to be deeply and systematically explored”. Given the major point raised above, this should be replaced with a more nuanced statement. Likewise, the example given in the Introduction of the fact that a denatured protein still has its catalytic groups and yet is inactive is trivial, and should be replaced by a more substantial discussion. The following paragraph, which starts with a discussion of X-ray structures, is an opportunity to fill in the reader on the extant work on residue coupling in proteins*.

To our knowledge this is the first quantitative analysis of an active site network of this extent, but we nevertheless agree with this and the above comments. Given the space limitations of the Abstract we have removed ‘deeply’ as also suggested above and said ‘yet to be quantitatively and systematically explored’ as our conclusions are derived from carrying out quantitative and systematic analyses.

*5) Again, given the major point raised above, the first paragraph of the paper goes from citations of historical work to citations of very general works on enzyme mechanism. The historical citations hardly seem relevant here, and it would be better to make the opening more substantial in terms of the proper intellectual context for this work*.

We have added the references and discussion suggested, as noted above, but we have also kept the more general historical perspective. Our experience is that this is quite helpful for the more general reader who is less familiar with these topics, to place them in a broader context. We believe that this is particularly appropriate for the *eLife* readership.